# SEL1L-HRD1 ER-associated degradation facilitates prohormone convertase 2 maturation and glucagon production in islet α cells

Wenzhen Zhu[1], Linxiu Pan [1,9], Xianwei Cui [2], Anna Chiara Russo[1], Rohit Ray [1], Brent Pederson[1,2], Xiaoqiong Wei [2,9], Liangguang Leo Lin [2,9], Mauricio Torres[2,9], Hannah Hafner [3], Brigid Gregg[3,4], Neha Shrestha[1,2], Chengyang Liu [5], Ali Naji[5], Peter Arvan [1,2], Darleen A. Sandoval [6,7], Iris Lindberg [8], Ling Qi [1,2,9] ✉ & Rachel Byerley Reinert [1] ✉

Proteolytic cleavage of proglucagon by prohormone convertase 2 (PC2) is required for islet α cells to generate glucagon. However, the regulatory mechanisms underlying this process remain largely unclear. Here, we report that SEL1L-HRD1 endoplasmic reticulum (ER)-associated degradation (ERAD), a highly conserved protein quality control system responsible for clearing misfolded proteins from the ER, plays a key role in glucagon production by regulating turnover of the nascent proform of the PC2 enzyme (proPC2). Using a mouse model with SEL1L deletion in proglucagon-expressing cells, we observe a progressive decline in stimulated glucagon secretion and a reduction in pancreatic glucagon content. Mechanistically, we find that endogenous proPC2 is a substrate of SEL1L-HRD1 ERAD, and that degradation of misfolded proPC2 ensures the maturation of activation-competent proPC2 protein in the ER. Here, we identify ERAD as a regulator of PC2 biology and an essential mechanism for maintaining α cell function.

Diabetes mellitus encompasses a spectrum of metabolic diseases with dysregulated secretion of insulin and glucagon from pancreatic islet β and α cells as a central feature. Interest in α cell biology has been reinvigorated in recent years, particularly with the advent of specific tools to manipulate α cell gene expression and glucagon signaling[1,2]. Hyperglucagonemia is observed in both type 1 and type 2 diabetes, contributing to hyperglycemia[3,4]. Glucagon receptor antagonists are in clinical development to improve hyperglycemia in diabetes[5,6], but may trigger dramatic α cell hyperplasia[7] like that observed in models with genetic deletion of glucagon[8] or its receptor (Gcgr)[9]. Conversely, recent studies have demonstrated a more positive role for α cells in directly promoting β cell function and survival through secretion of proglucagon-derived peptides such as glucagon and/or glucagon-like peptide 1 (GLP-1) and their interaction with GCGR and GLP1R receptors

[1]Division of Metabolism, Endocrinology & Diabetes, Department of Internal Medicine, University of Michigan Medical School, Ann Arbor, MI, USA. [2]Department of Molecular & Integrative Physiology, University of Michigan Medical School, Ann Arbor, MI, USA. [3]Department of Pediatrics, Division of Pediatric Endocrinology, University of Michigan, Ann Arbor, MI, USA. [4]Department of Nutritional Sciences, School of Public Health, University of Michigan, Ann Arbor, MI, USA. [5]Department of Surgery, Perelman School of Medicine, University of Pennsylvania, Philadelphia, PA, USA. [6]Department of Surgery, University of Michigan, Ann Arbor, MI, USA. [7]Department of Pediatrics, Nutrition Section, University of Colorado Anschutz Medical Campus, Aurora, CO, USA. [8]Department of Anatomy and Neurobiology, University of Maryland-Baltimore, Baltimore, MD, USA. [9]Present address: Department of Molecular Physiology and Biological Physics, University of Virginia School of Medicine, Charlottesville, VA, USA. ✉e-mail: xvr2hm@virginia.edu; reinertr@med.umich.edu

on β cells[10–12]. The cellular mechanisms that define the ability of α cells to regulate the production and maturation of proglucagon-derived peptides remains an active area of investigation.

In α cells, proglucagon undergoes limited proteolysis primarily by prohormone convertase 2 (PC2), which generates glucagon as the major peptide product[13]. PC2 is initially synthesized and folded as a 75 kDa inactive zymogen, proPC2, in the endoplasmic reticulum (ER). The maturation of proPC2 to its ~64 kDa active form, PC2, is a complex, multi-step process that requires folding in the ER, binding to its chaperone protein 7B2, posttranslational modification in the Golgi network, and sorting into acidic secretory granules, where it is then activated by intramolecular autocatalysis[14,15]. Understanding the regulatory mechanisms underlying proPC2 maturation and processing is essential for optimizing glucagon production and developing targeted therapeutic strategies for manipulating α cell function.

The synthesis and packaging of islet hormones and their processing enzymes into secretory granules depends on effective protein quality systems, particularly within the ER. Genome-wide association studies (reviewed in ref. 16) and multi-omic analyses of human islets[17] have linked genes involved in ER homeostasis and ER stress responses with both type 1 and type 2 diabetes, but many of these underlying mechanisms have primarily been studied in β cells[18,19]. The specific mechanisms defining ER homeostasis in α cells are poorly understood. As the initial location of protein synthesis, the ER must maintain a healthy environment for proteins to fold into their mature structure, as failure to do so risks pathologic aggregation or premature activation. Cells employ multiple, interrelated quality control mechanisms to respond to the ever-changing burden of protein synthesis and manage protein misfolding, including the unfolded protein response (UPR), autophagy, and ER-associated degradation (ERAD)[20].

ERAD is a key quality control mechanism responsible for targeting misfolded proteins in the ER for cytosolic proteasomal degradation[21,22]. The SEL1L-HRD1 protein complex represents the most conserved branch of ERAD, where SEL1L serves as a cognate cofactor for the E3 ligase HRD1[23]. We and others have shown that SEL1L controls both HRD1 protein stability[24,25] and ERAD complex formation[26,27]. SEL1L-HRD1 ERAD promotes the maturation of several misfolding-prone prohormones, such as pro-arginine vasopressin (proAVP) and pro-opiomelanocortin (POMC) in neurons[28,29]. In the absence of SEL1L, proAVP and POMC peptides aggregate in the ER, impairing their maturation and limiting production of their functional hormone derivatives. In pancreatic islet β cells, SEL1L-HRD1 ERAD is critical to maintain cell identity[30,31] and insulin production[32,33]. The role of SEL1L-HRD1 ERAD in islet α cells and proglucagon maturation has until now been unexplored.

In this study, we generated a mouse model inactivating SEL1L in proglucagon-expressing cells and found that defective ERAD is associated with impaired glucagon production secondary to impaired maturation of proPC2. Mechanistically, we uncovered an unexpected role of SEL1L-HRD1 ERAD in the quality control of proPC2 in the ER – ensuring that misfolded proPC2 is targeted for degradation, thereby maintaining the integrity of proglucagon processing machinery. In the absence of proper ERAD function, misfolded proPC2 accumulates in the ER, undergoes aberrant proteolytic cleavage and forms high molecular weight disulfide-bonded complexes, leading to impaired maturation and reduced levels of active PC2. This, in turn, limits the cleavage of proglucagon, resulting in diminished glucagon production and secretion.

## Results

### The ER network and SEL1L-HRD1 ERAD expression in islet α cells

Like islet β cells[30,31], neighboring α cells also have an expansive network of ER, as observed by transmission electron microscopy (TEM) (Fig. 1a, and Supplementary Fig. 1a). To explore the role of ER homeostasis in α cells, we examined the ER in glucagon-null Gcg-STOP-flox ("Gcg−/−")

mice[34]. In this model, there is a dramatic expansion of the α cell population secondary to hyperaminoacidemia, due to the complete absence of hepatic glucagon signaling[35,36]. Despite their inability to make proglucagon and its derivative peptides, the ER remained prominent in Gcg−/− α cells (Fig. 1b, and Supplementary Fig. 1a-b). Using confocal microscopy, we found that protein levels of BiP, an ER chaperone protein, is relatively low in normal α cells compared to that in β cells in wild-type mice (Fig. 1c, d). However, BiP was increased in α cells in Gcg−/− mice (Fig. 1c, d). This expansion of ER capacity in Gcg−/− α cells was also reflected by increased expression of SEL1L compared to that in normal α cells (Fig. 1e, f, white arrows). SEL1L was expressed in both α and β cells in human islets, from donors with or without diabetes (Fig. 1g). These findings led us to investigate the role of SEL1L-HRD1 ERAD in α cell ER homeostasis and function.

### Targeted deletion of SEL1L in islet α cells

To define the role of SEL1L-HRD1 ERAD in α cells, we developed a mouse model in which SEL1L was inactivated specifically in proglucagon-expressing cells by crossing mice expressing an "improved" constitutively active Cre recombinase under the proglucagon promoter (Gcg^iCre)[37] with Sel1L-floxed mice on a B6 background[24]. In these Sel1L^ΔGcg mice, almost all α cells showed absence of immunolabeling for SEL1L (Fig. 2a). Confirming that ERAD function was impaired in Sel1L^ΔGcg α cells, we also observed increased expression of the ERAD substrate OS9 (Fig. 2b) and the ER chaperone BiP (Fig. 2c). By contrast, immunofluorescence of glucagon was slightly reduced in Sel1L^ΔGcg α cells (Fig. 2d), suggesting that ERAD impairs glucagon production. However, this appears to occur through an indirect mechanism, as ERAD dysfunction did not lead to retention of proglucagon in the ER (Fig. 2d), as has been shown with other misfolding-prone prohormones such as proAVP and POMC[28,29].

Despite targeting SEL1L deletion in all proglucagon-expressing cells in Sel1L^ΔGcg mice, we did not observe increased BiP expression in intestinal L-cells (Supplementary Fig. 2a), as detected by an anti-glucagon antibody that shows colocalization with anti-GLP-1 antibody (Supplementary Fig. 2b) and also detects proglucagon by Western blotting (Supplementary Fig. 1c; antibody details provided in Supplementary Table 1). To better understand the extent of Cre recombinase activity in this model, we generated R26R-EYFP^ΔGcg reporter mice by crossing ROSA26-STOP-flox-EYFP mice[38] with the same Gcg^iCre line[37]. We found that the vast majority of islet α cells expressed the YFP reporter (Supplementary Fig. 2c), while YFP was not detected in intestinal L-cells (Supplementary Fig. 2d). In line with this notion, there was no difference in glucose-stimulated total GLP-1 levels in serum (Supplementary Fig. 2f) or in GLP-1 content of distal colon epithelium (Supplementary Fig. 2g) from Sel1L^ΔGcg mice. Together, these data suggest that this Gcg^iCre line predominantly targets islet α cells.

### Attenuated glucagon production Sel1L^ΔGcg mice

We next asked how inactivation of SEL1L-HRD1 ERAD affected α cell function. Male and female Sel1L^ΔGcg mice showed normal growth on chow diet through adulthood (Fig. 3a). There was no change in oral glucose tolerance (Fig. 3b) in 4−8 month-old Sel1L^ΔGcg mice compared to sex-matched littermate controls. To assess stimulated glucagon secretion in vivo, we subjected adult cohorts to insulin-induced hypoglycemia. At 4-8 months of age, male Sel1L^ΔGcg mice had significantly decreased glucagon secretion following hypoglycemia (Fig. 3c). Female Sel1L^ΔGcg mice also showed reduced glucagon secretion in vivo, but starting at 8 months of age (Fig. 3c).

To understand the mechanism underlying lower glucagon levels in Sel1L^ΔGcg mice, we next investigated glucagon production and secretion in α cells. In both adult male and female Sel1L^ΔGcg mice, pancreatic glucagon content was reduced by half by 4 months of age (Fig. 3d), as detected by an ELISA assay targeted to detect the N- and C-terminal ends of the mature 29-amino acid glucagon peptide. This

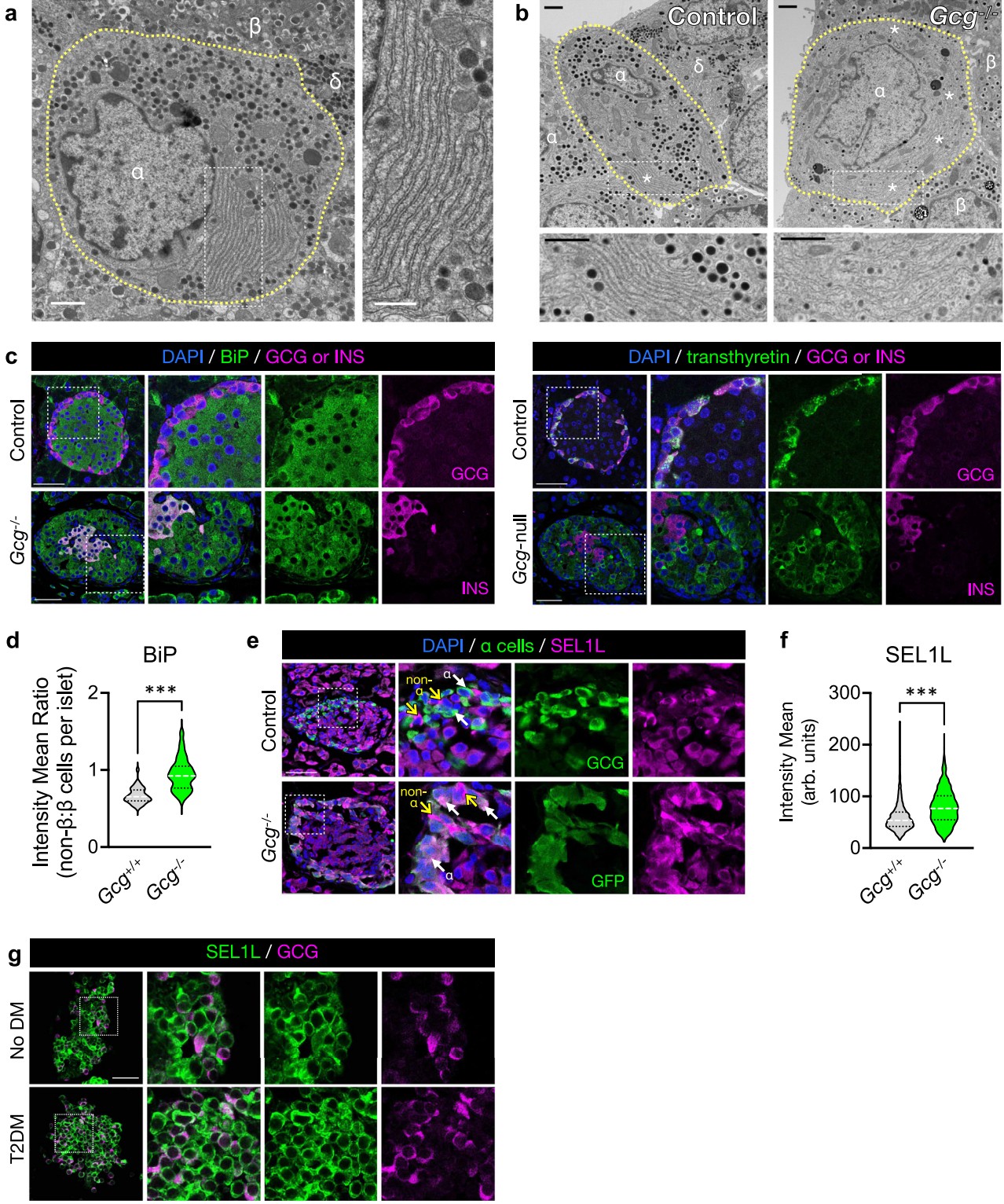

did not appear to reflect a developmental defect in α cell formation, as stimulated glucagon secretion was not impaired in 11-week-old *Sel1L^{ΔGcg}* mice (Supplementary Fig. 2h) and pancreatic glucagon content was reduced only in male mice at that age (Supplementary Fig. 2i). The defect in circulating glucagon levels was not due to defects in secretory dynamics or response to stimuli, as isolated islets from *Sel1L^{ΔGcg}* mice showed a similar secretory response to low glucose with or without the secretagogue L-arginine, when normalized to total glucagon content (Fig. 3e). Importantly, the expression of Cre in the native *Gcg* locus did not affect pancreatic glucagon content in *R26R-EYFP^{ΔGcg}* reporter mice (Supplementary Fig. 2e). Altogether, these data suggest that ERAD dysfunction limits glucagon production in α cells, while having no effect on glucagon secretory dynamics.

## ER dilation in *Sel1L^{ΔGcg}* α cells

We next performed TEM to examine α cell morphology and secretory granule formation. In contrast to the prominent stacks of ER and abundant electron-dense granules in normal α cells, the α cells in

**Fig. 1 | The endoplasmic reticulum (ER) is a prominent feature of normal and proliferative islet α cells. a** Transmission electron microscopy (TEM) image of a normal α cell (yellow dashed outline), demonstrating prominent stacks of rough ER. Adjacent β and δ cells are labeled. Scale bar of left panel, 1 μm; of inset, 600 nm. **b** TEM images showing prominent ER stacks (asterisks) in highly proliferative α cells from *Gcg⁻/⁻* mice compared to littermate controls. Scale bars, 1 μm. Additional TEM images are shown in Supplementary Fig. 1a. **c** Representative islets from *Gcg⁻/⁻* mice and littermate controls, with adjacent tissue sections immunolabeled for the ER chaperone BiP (left) or the α cell marker transthyretin (right) and either glucagon (GCG, top panels; BMA Biomedicals) or insulin (INS, bottom panels). **d** Violin plot shows the distribution of relative BiP immunofluorescence in islets from *Gcg⁻/⁻* and control mice, quantified as the ratio of the average BiP intensity of non-β cells to β cells, expressed as arbitrary units (arb. units). Data represent a total of 37 islets from *n* = 3 *Gcg⁺/⁺* mice and 61 islets from *n* = 6 *Gcg⁻/⁻* mice, ***P* < 0.001, unpaired two-tailed Student's t test. **e** Representative islet cryosections immunolabeled for the ER-associated degradation (ERAD) component SEL1L and either glucagon (GCG, top panels) or GFP (as expressed by the construct inserted into the *Gcg* gene, bottom panels). White arrows mark examples of α cells. Yellow arrows denote peripheral non-α islet cells that highly express SEL1L, notably with absence of glucagon expression. **f** Violin plot shows the distribution of SEL1L immunofluorescence in GCG+ or GFP+ α cells, normalized to the mean intensity of SEL1L in β cells within the same islet, expressed as arbitrary units (arb. units). Data represent a total of 1476 α cells from *n* = 3 *Gcg⁺/⁺* mice and 2348 α cells from *n* = 3 *Gcg⁻/⁻* mice, ***P* < 0.001, unpaired two-tailed Student's t test. **g** Cryosections of isolated islets from human islet donors immunolabeled for SEL1L and glucagon (GCG). The islet in the top panels is from a donor without diabetes (No DM), and the islet in the bottom panels is from a donor with a two-year history of type 2 diabetes (T2DM); see Supplementary Table 2 for further donor characteristics. Scale bar for left panels in (**c**, **e**, **g**) 50 μm. Specific *P* values are available in the source data provided as a Source Data file.

*Sel1L^ΔGcg* mice showed significant ER dilation and, in extreme cases, fewer glucagon granules (Fig. 3f and Supplementary Fig. 3a). Nonetheless, the glucagon granules in *Sel1L^ΔGcg* α cells appeared largely normal in morphology (Fig. 3f and Supplementary Fig. 3a). Additionally, β cell architecture and pancreatic insulin content were unchanged in *Sel1L^ΔGcg* mice (Supplementary Fig. 3b, c). By immunofluorescence detection with antibodies to the mature glucagon sequence, α cell mass was reduced in adult *Sel1L^ΔGcg* mice (Supplementary Fig. 3d), but without significant changes in α cell apoptosis (Supplementary Fig. 3e) or proliferation (Supplementary Fig. 3f), as evaluated by TUNEL assay and Ki67+ cells, respectively. Altogether, these data suggest that ERAD deficiency causes ER dilation and limits production of mature glucagon in α cells.

## ERAD deficiency causes the accumulation of proPC2 protein in vivo

Given the greater reduction in mature glucagon relative to proglucagon levels in *Sel1L^ΔGcg* α cells, we speculated that the defect may lie in the proteolytic cleavage of proglucagon, a process catalyzed by the PC2 enzyme. PC2 is initially synthesized in the ER as the zymogen proPC2. ProPC2 folds in the ER, forming three disulfide bonds (Fig. 4a), binds its chaperone 7B2 to proceed through the secretory pathway[39], and then becomes activated within secretory granules[40], where the Pro domain is cleaved from the proprotein by autocatalysis. AlphaFold structure modeling was used to visualize the possible conformation of each domain (Fig. 4b). This tertiary structure aligns with prior work showing that proPC2 is susceptible to misfolding and aggregation within the ER in the absence of its chaperone protein 7B2[39,41]. We thus asked how SEL1L-HRD1 ERAD affects PC2 biology.

To visualize the expression level and distribution of proPC2 in vivo, we performed immunofluorescence labeling using antibodies specific for different regions of proPC2[39] (Fig. 4a). There was a marked increase in proPC2 levels in *Sel1L^ΔGcg* α cells (Fig. 4c). In line with this finding, immunolabeling with antibodies to the catalytic domain of PC2 (which detects both mature PC2 and the proPC2 protein) showed that total PC2 levels were also increased in *Sel1L^ΔGcg* α cells (Fig. 4d). These data suggest that SEL1L deficiency causes the accumulation of proPC2 protein in vivo.

## Altered proPC2 processing in the absence of SEL1L-HRD1 ERAD

To further explore how SEL1L-HRD1 ERAD regulates proPC2 maturation in α cells, we developed a cell culture model of ERAD deficiency using CRISPR-mediated deletion of either SEL1L or HRD1 in the well-established αTC1-6 α cell line (ΔSel1L and ΔHrd1, respectively) (Fig. 5a, and Supplementary Fig. 4a–c). Deletion of SEL1L or HRD1 in αTC1-6 cells caused ERAD dysfunction, as demonstrated by increased protein levels of ERAD substrates IRE1α and OS9 (Supplementary Fig. 4d–f). In addition, ERAD deficiency triggered a mild activation of ER stress

responses as measured by the phosphorylation of eIF2α and splicing of *Xbp1* mRNA (Supplementary Fig. 4g–k). Compared to cells treated with the ER stressor tunicamycin, the level of *Xbp1* mRNA splicing in ERAD-deficient αTC1-6 cells was modest (Supplementary Fig. 4k), similar to our previous observations[29,30].

We then examined (pro)PC2 expression using Western blot with three different antibodies targeted to different regions of the protein (Fig. 4a). Using a proPC2-specific antibody, we found that expression of proPC2 (~75 kDa) was significantly increased in both ΔSel1L and ΔHrd1 α cells compared to control cells (Fig. 5b). Intriguingly, we observed a novel ~55 kDa proPC2 fragment (which we termed proPC2*) that was highly enriched in ERAD-deficient cells (Fig. 5b). In addition, using an antibody targeting the PC2 catalytic domain, we confirmed the accumulation of proPC2 and proPC2* in both ΔSel1L and ΔHrd1 α cells compared to control cells, while mature PC2 (~64 kDa) protein levels were reduced by half (Fig. 5c). The combined increase in full-length proPC2 and the proPC2* fragment led to a slight increase in the overall expression of proPC2-derived proteins in ΔSel1L α cells (Supplementary Fig. 5a, b), reflecting the pattern observed in *Sel1L^ΔGcg* islets. We confirmed that proPC2* expression was also increased in *Sel1L^ΔGcg* islets by Western blot (Supplementary Fig. 5c). In contrast, using an antibody specific for the C-terminal domain of PC2, we failed to observe proPC2* in ΔSel1L or ΔHrd1 α cells (Fig. 5d), suggesting that proPC2* was generated by an aberrant cleavage at the C-terminal end of the protein. We confirmed that proPC2* was derived from proPC2 given its absence in cells with CRISPR-mediated co-deletion of PC2 and Sel1L (Fig. 5e, f; note that PC2 was significantly reduced but incompletely deleted by this method). We also confirmed that proPC2* was an immature protein found in the ER, given its sensitivity to deglycosylation following endoglycosidase H or PNGase F treatments (Supplementary Fig. 5d).

To explore whether the consequences of ERAD deficiency specifically affected proPC2 maturation and glucagon production in α cells, we next examined expression of the PC1/3 enzyme, which is primarily responsible for generation of GLP-1 from proglucagon. By immunofluorescence, *Sel1L^ΔGcg* α cells showed slightly reduced expression of both PC1/3 and GLP-1 (Supplementary Fig. 6a, b). Unlike proPC2, we did not observe aggregation of PC1/3 under nonreducing conditions on Western blot (Supplementary Fig. 6c). Instead, the lower PC1/3 content was attributed in part to reduced expression of *Pcsk1* mRNA in ΔSel1L α cells (Supplementary Fig. 6d). Pancreatic total GLP-1 content, as measured by immunoassay, was significantly reduced in adult male but not female *Sel1L^ΔGcg* mice (Supplementary Fig. 6e). Thus, ERAD deficiency specifically inhibits maturation of proPC2 in α cells and imparts a lesser effect on PC1/3 expression and GLP-1 production.

The accumulation of both proPC2 and proPC2* was specific to ERAD deficiency, and uncoupled from ER stress, as it was not observed in control αTC1-6 cells following induction of ER stress with

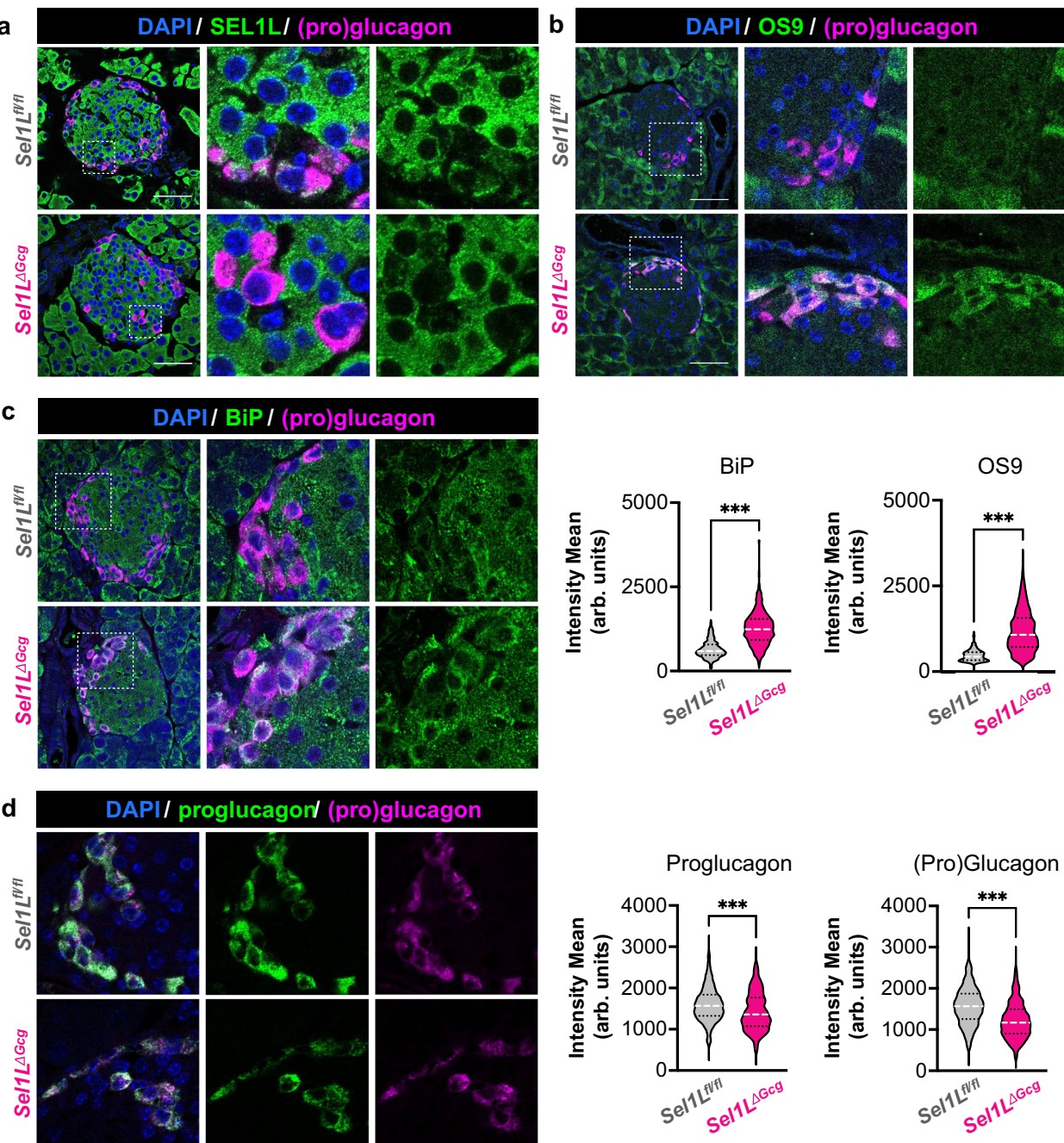

**Fig. 2 | Inactivation of SEL1L-HRD1 ERAD disrupts ER homeostasis in α cells but does not lead to accumulation of proglucagon-derived peptides.** Representative islets from adult (4–8 month-old) *Sel1L^ΔGcg* mice and *Sel1L^fl/fl* littermate controls, immunolabeled for (pro)glucagon and the ERAD component SEL1L (**a**) the ERAD cofactor/substrate OS9 (**b**) the ER chaperone BiP (**c**) or proglucagon (**d**). Note that the BMA Biomedicals antibody targeting the mature glucagon sequence was found to label full-length proglucagon by Western blot (see Supplementary Fig. 1c), so images are labeled as detecting "(pro)glucagon" accordingly. Scale bars of left panels, 50 μm. Violin plots show the distribution of immunofluorescence intensity in α cells for each protein, expressed as arbitrary units (arb. units). For BiP, 564 α cells were evaluated in *n* = 5 *Sel1L^fl/fl* mice and 1361 cells in *n* = 5 *Sel1L^ΔGcg* mice; for OS9, 277 α cells were evaluated in *n* = 3 *Sel1L^fl/fl* mice and 874 cells in *n* = 3 *Sel1L^ΔGcg* mice; for proglucagon, 892 α cells were evaluated in *n* = 5 *Sel1L^fl/fl* mice and 510 cells in *n* = 5 *Sel1L^ΔGcg* mice; for (pro)glucagon, 1451 α cells were evaluated in *n* = 4 *Sel1L^fl/fl* mice and 936 cells in *n* = 4 *Sel1L^ΔGcg* mice. \*\*\**P* < 0.001, unpaired two-tailed Student's t test. Specific *P* values are available in the source data provided as a Source Data file.

either tunicamycin or thapsigargin (Fig. 5g). Importantly, ERAD deficiency was associated with a slight decrease, not increase, in expression of *Pcsk2* mRNA (Fig. 5h). Therefore, the accumulation of proPC2 protein was attributed to post-transcriptional regulation. Taken together, these data show that SEL1L-HRD1 ERAD is critical for the maturation of proPC2 in α cells by preventing its abnormal proteolytic cleavage.

**Nascent proPC2 is degraded by SEL1L-HRD1 ERAD in α cells**

We then explored the molecular mechanism underlying the interplay between ERAD and proPC2. Using cycloheximide to block protein synthesis, we found that both proPC2 and proPC2\* were stabilized in both ΔSel1L and ΔHrd1 α cells, as validated by two different antibodies (Fig. 6a, b). Moreover, in the absence of ERAD, proPC2 formed high molecular weight (HMW) protein complexes as visualized by non-

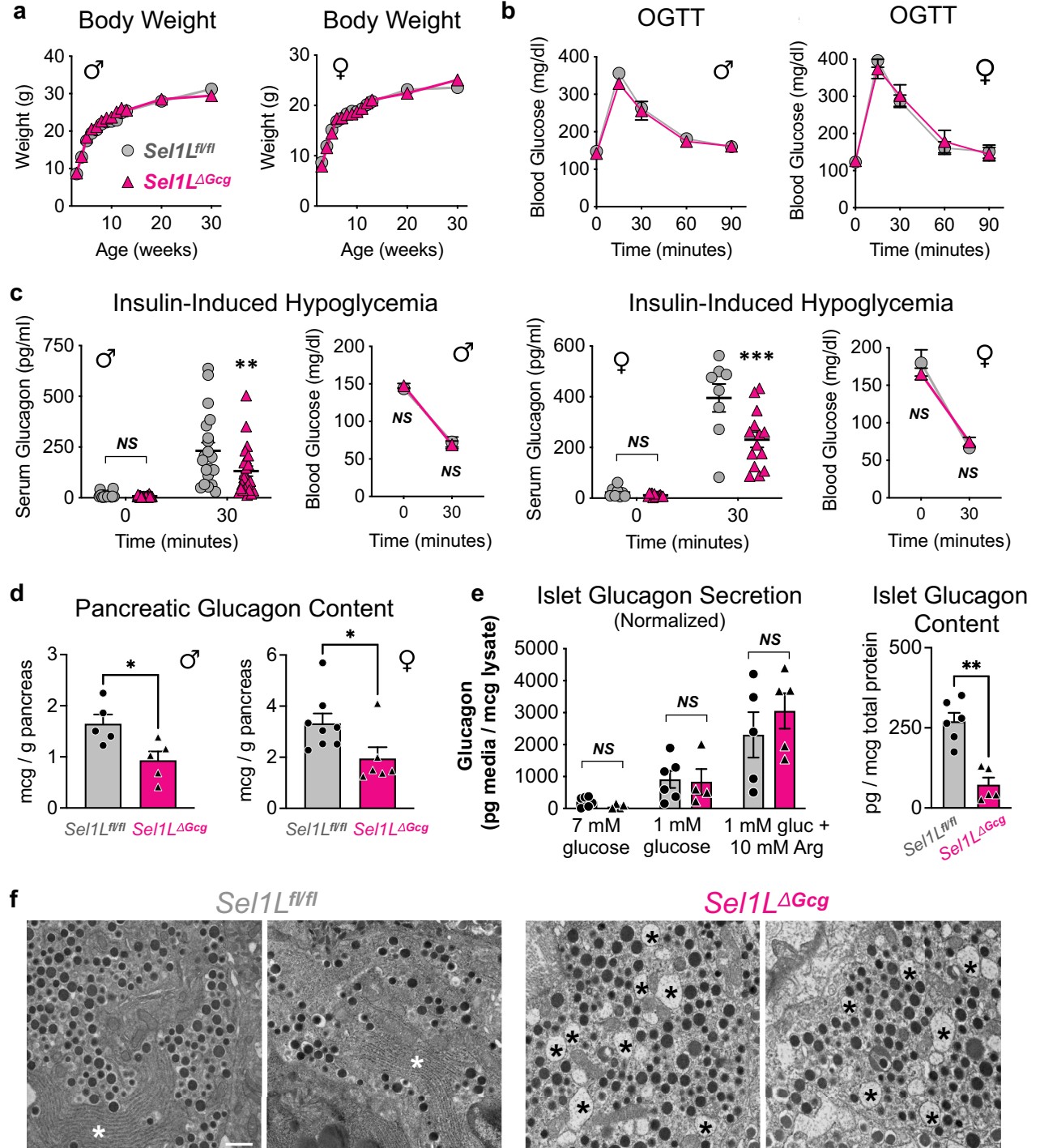

**Fig. 3 | ERAD inactivation in α cells does not affect systemic growth or glucose tolerance but limits glucagon production with age. a, b** Body weight and oral glucose tolerance (OGTT) in adult (4–8 month-old) male and female *Sel1L^ΔGcg* mice (triangles) and *Sel1L^fl/fl* controls (circles). Each data point in a represents *n* > 6 mice, shown as mean ± SEM. In b, *n* = 9 male *Sel1L^fl/fl* mice and *n* = 6 male *Sel1L^ΔGcg* mice; *n* = 6 female *Sel1L^fl/fl* mice and *n* = 7 female *Sel1L^ΔGcg* mice, shown as mean ± SEM. **c** Serum glucagon and glucose levels in vivo in adult male (>4 months) and female (>8 months) *Sel1L^ΔGcg* mice (triangles) and *Sel1L^fl/fl* controls (circles) in response to insulin-induced hypoglycemia. For serum glucagon, each data point represents data from one mouse, with mean ± SEM bars. For males, *n* = 23 *Sel1L^fl/fl* mice and *n* = 31 *Sel1L^ΔGcg* mice; for females, *n* = 8 *Sel1L^fl/fl* mice and *n* = 14 *Sel1L^ΔGcg* mice. For blood glucose, data points represent group mean ± SEM. **d** Acid ethanol-extracted pancreatic glucagon content in adult (4–8 month-old) mice. Each data point

represents data from one mouse, with bars showing mean ± SEM. For males, *n* = 5 *Sel1L^fl/fl* mice and *n* = 5 *Sel1L^ΔGcg* mice; for females, *n* = 8 *Sel1L^fl/fl* mice and *n* = 6 *Sel1L^ΔGcg* mice. **e** Normalized glucagon secretion and total islet glucagon content from isolated *Sel1L^ΔGcg* and *Sel1L^fl/fl* islets. Each data point represents data from one mouse (*n* = 6 *Sel1L^fl/fl* and *n* = 5 *Sel1L^ΔGcg*), with mean ± SEM bars. **f** Representative transmission electron microscopy images of α cells from adult *Sel1L^ΔGcg* mice and *Sel1L^fl/fl* controls, with ER stacks or dilated ER lumen denoted by asterisks. Scale bar, 600 nm. For statistical tests, *P < 0.05, **P < 0.01, ***P < 0.001; *NS*, not significant with *P* > 0.05; a two-way ANOVA with Šidák post-test was performed for serum glucagon tests in (**c**) and an unpaired two-tailed Student's t test was performed to compare groups in all other graphs. Full data and specific *P* values are available in the source data provided as a Source Data file.

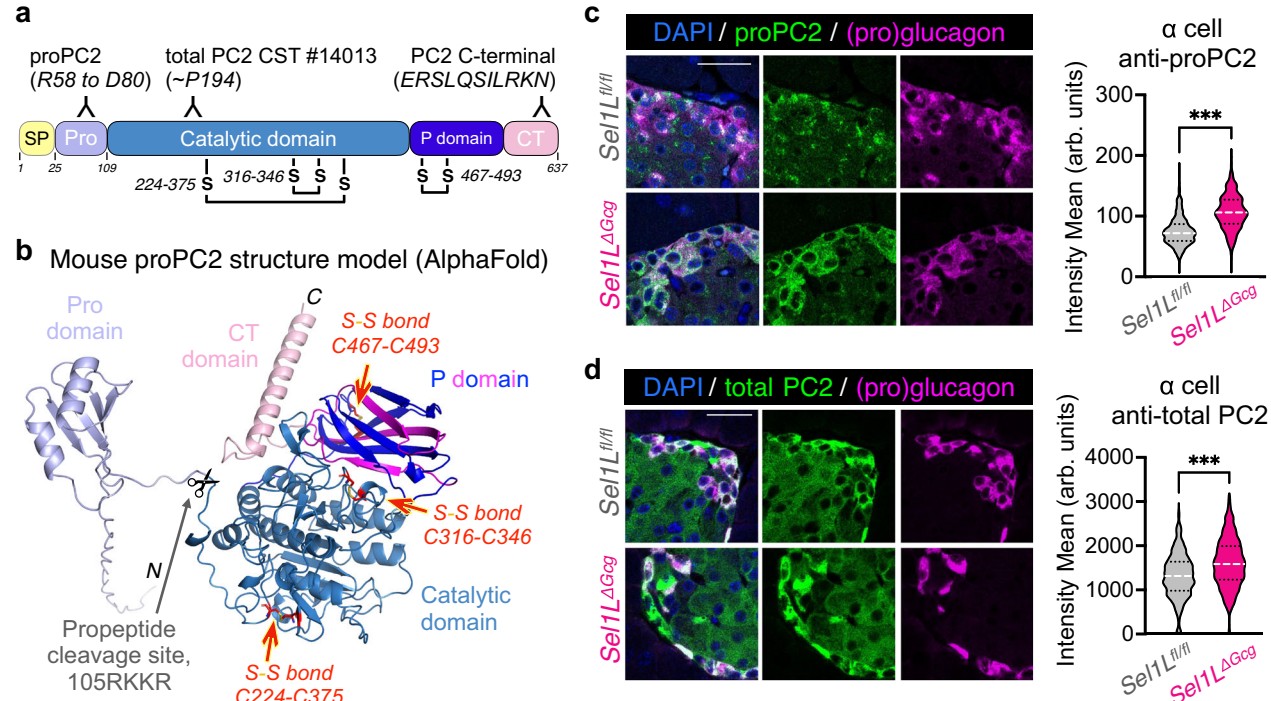

**Fig. 4 | Inactivation of SEL1L-HRD1 ERAD in α cells leads to accumulation of prohormone convertase 2 in vivo. a** Schematic diagram of the mouse preproPC2 protein, denoting the locations of antibody targets. SP: signal peptide, Pro: propeptide, CT: C-terminal region. **b** AlphaFold 3 structural model of the mouse proPC2 protein. **c, d** Representative islets from *Sel1L*[ΔGcg] mice and *Sel1L*[fl/fl] controls, immunolabeled for the PC2 proprotein (**c**) or total prohormone convertase 2 (PC2 CST, **d**). Scale bars, 25 μm. Violin plots at right show the distribution of immunofluorescence intensity of (pro)PC2 in glucagon+ cells, expressed as arbitrary units (arb. units). Data for proPC2 were obtained from 1151 α cells from *n* = 3 *Sel1L*[fl/fl] mice and 1331 α cells from *n* = 4 *Sel1L*[ΔGcg] mice, and data for PC2 were obtained from 1044 α cells from *n* = 4 *Sel1L*[fl/fl] mice and 840 α cells from *n* = 4 *Sel1L*[ΔGcg] mice. ***$P$ < 0.001; unpaired two-tailed Student's *t* test. Specific $P$ values are available in the source data provided as a Source Data file.

reducing SDS-PAGE. The formation of these HMW complex were mediated by aberrant disulfide bonds as they were sensitive to the reducing agent β-mercaptoethanol (lanes 5-6, Fig. 6c). The proPC2* fragment contributed to this process, as it co-migrated with HMW aggregates under non-reducing conditions (lanes 5-6 versus lanes 2-3, Fig. 6c). Supporting the direct role of SEL1L-HRD1 ERAD in regulating proPC2 degradation, we found that proPC2 was polyubiquitinated in a HRD1-dependent manner (Fig. 6d). MG132 treatment also increased the polyubiquitination of proPC2 (lanes 4-5, Fig. 6d), pointing to the involvement of proteasome. Interestingly, MG132 treatment also increased the production of proPC2* in control cells (input lane 4 versus input lanes 1-2, Fig. 6d). This suggests that proPC2* is routinely generated in normal α cells but is undetectable due to degradation by SEL1L-HRD1 ERAD. These data demonstrate that in the absence of SEL1L-HRD1 ERAD, proPC2 accumulates in the α cell ER, is abnormally cleaved, and forms HMW complexes via aberrant disulfide bonds.

To better understand the timing of proPC2 processing defects following ERAD dysfunction, we used an siRNA approach to acutely knock down SEL1L or HRD1 protein levels in control αTC cells (Supplementary Fig. 7). Similar to CRISPR-mediated inactivation of ERAD proteins in αTC cells, we observed accumulation of proPC2 and proPC2* 48 h after siRNA knockdown of these ERAD proteins (Supplementary Fig. 7a). We also observed accumulation of full-length proglucagon in siSel1L and siHrd1 cells (Supplementary Fig. 7b). Notably, we confirmed the profound accumulation of proglucagon and absence of mature glucagon formation in CRISPR-generated ΔPC2 cells, in which complete inactivation of functional PC2 disrupts proglucagon-to-glucagon conversion, as previously shown[42]. These data further support the role of SEL1L-HRD1 ERAD in proPC2

maturation and subsequent PC2-mediated conversion of proglucagon into the mature glucagon peptide.

## Impaired proPC2 enzymatic activity in ERAD-deficient α cells

One unique feature of PC2 is its requirement for a protein cofactor, 7B2 (also called secretogranin-5), to serve as a chaperone that binds proPC2 in the ER and inhibits PC2 enzyme activation until it reaches later stages in the secretory pathway[39]. In the absence of 7B2, proPC2 is prone to aggregation[41], thus limiting generation of glucagon from proglucagon[43]. We found that 7B2 protein expression was reduced in both ΔSel1L and ΔHrd1 α cells compared to their respective controls (Fig. 6e), suggesting a possible contribution to proPC2 misprocessing. To better understand the relationship between 7B2 levels and proPC2 maturation in ERAD deficiency, we investigated 7B2 expression in αTC cells following siRNA-mediated ERAD inactivation. In contrast to ΔSel1L and ΔHrd1 αTC cells with chronic ERAD inactivation, both siSel1L and siHrd1 cells instead showed an increase in 7B2 expression (Supplementary Fig. 7a), suggesting that ERAD function influences 7B2 expression, but independently from its effect on proPC2. Importantly, overexpression of 7B2 was unable to rescue the accumulation and aggregation of proPC2(*) in ΔSel1L and ΔHrd1 αTC cells (Supplementary Fig. 8a, b), supporting a limited role for 7B2 in the proPC2 misprocessing observed in ERAD deficiency.

To determine whether these defects in proPC2 maturation affected PC2 enzymatic function, we measured PC2 activity in αTC cells using a substrate-specific aminomethylcoumarin assay as previously described[44]. This assay detects activity of mature PC2 in addition to properly folded proPC2 that is spontaneously autoactivated in the low pH of the reaction buffer[15]. Compared to control cells, ΔSel1L and ΔHrd1 αTC cells showed significantly impaired PC2 activity (Fig. 6f).

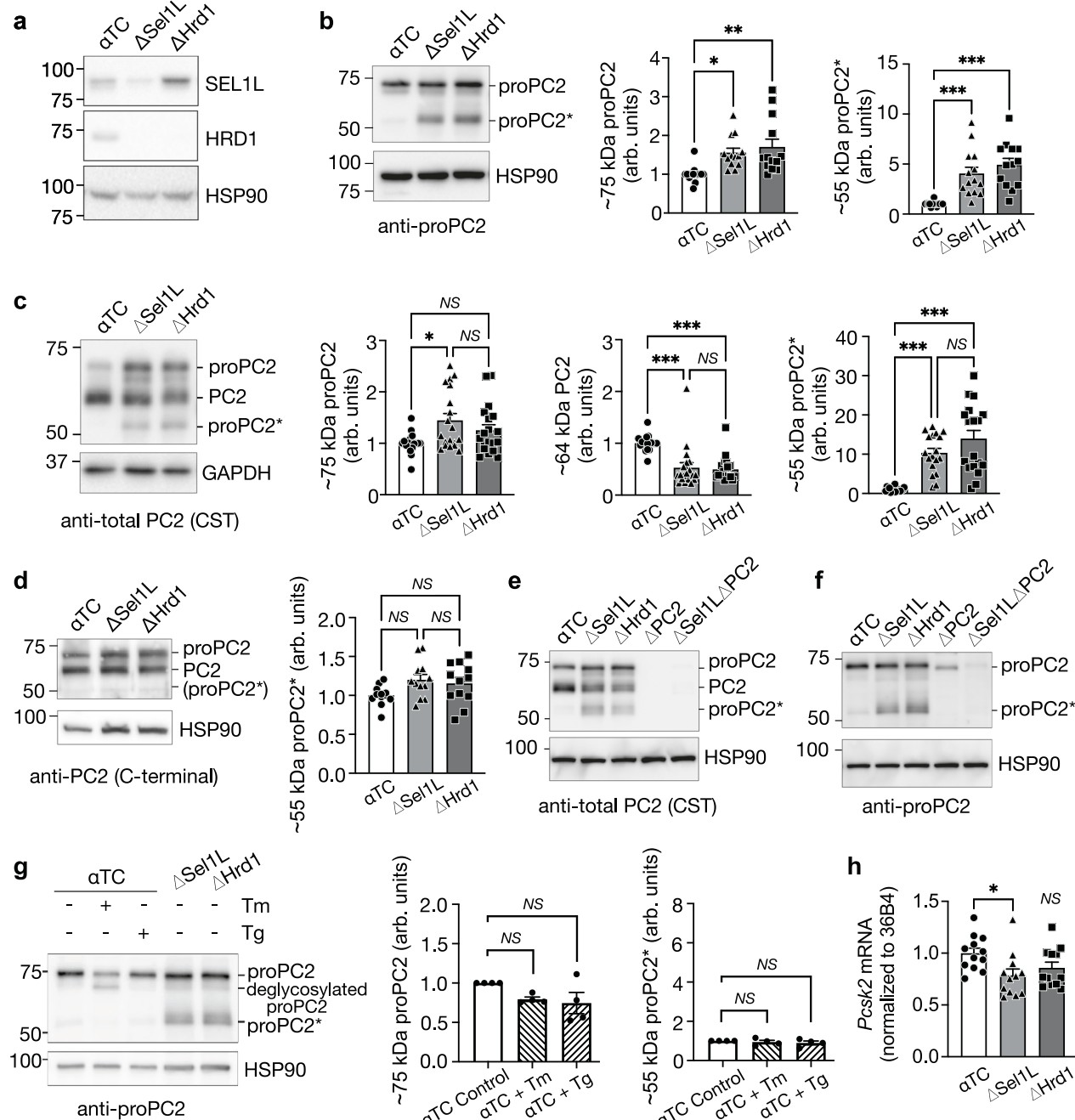

**Fig. 5 | Inactivation of SEL1L-HRD1 ERAD in α cells alters processing of pro-hormone convertase 2. a** Western blot of SEL1L and HRD1 expression in αTC1-6 cells following CRISPR-mediated deletion of SEL1L (ΔSel1L) or HRD1 (ΔHrd1) compared to vector-treated controls (αTC). Quantification is shown in Supplementary Fig. 4b, c. **b–d** Western blots of αTC cells using antibodies to the PC2 proprotein (**b**) PC2 catalytic domain (**c**) or PC2 C-terminal domain (**d**) with band quantification shown at right. Note the generation of an N-terminal fragment of the proPC2 protein in ΔSel1L and ΔHrd1 cells, denoted proPC2*. **e, f** Western blot of αTC cells following CRISPR-mediated deletion of SEL1L (ΔSel1L), HRD1 (ΔHrd1), and/or PC2 (ΔPC2; note that incomplete PC2 deletion was observed), labeled with antibodies to the catalytic domain of PC2 (**e**) or to the PC2 proprotein (**f**). **g** Western

blot of tunicamycin (Tm)- and thapsigargin (Tg)-treated control αTC cells, labeled with an antibody to proPC2, with band quantification shown at right. **h** Quantitative RT-PCR of the *Pcsk2* gene encoding PC2 in αTC cells. Note that molecular weight markers are listed in kDa in **a–g**. In graphs, ΔSel1L samples are represented by light gray/triangles, ΔHrd1 represented by dark gray/squares, and αTC controls represented by white/circles, unless treated with Tm or Tg as otherwise noted. In bar graphs, each data point represents a single biological replicate from 2-3 independent experiments, shown as mean ± SEM, expressed as arbitrary units (arb. units). *$P < 0.05$; **$P < 0.01$; ***$P < 0.001$; *NS*, not significant ($P > 0.05$); one-way ANOVA with Šidák post-test. For space limitations, specific *P* values and *n* numbers for independent experiments are available in the source data provided as a Source Data file.

Addition of the 7B2 C-terminal peptide, a potent and specific PC2 inhibitor[44], successfully abolished PC2 activity in all cells as a control (Fig. 6f). Together, these data suggest that the aberrant cleavage and aggregation of proPC2 in ERAD-deficient α cells impairs PC2 activity.

## Discussion

In this study, we report SEL1L-HRD1 ERAD as a direct regulator of islet α cell function. We found that SEL1L-HRD1 ERAD is essential for efficient glucagon biogenesis by facilitating the folding and maturation of

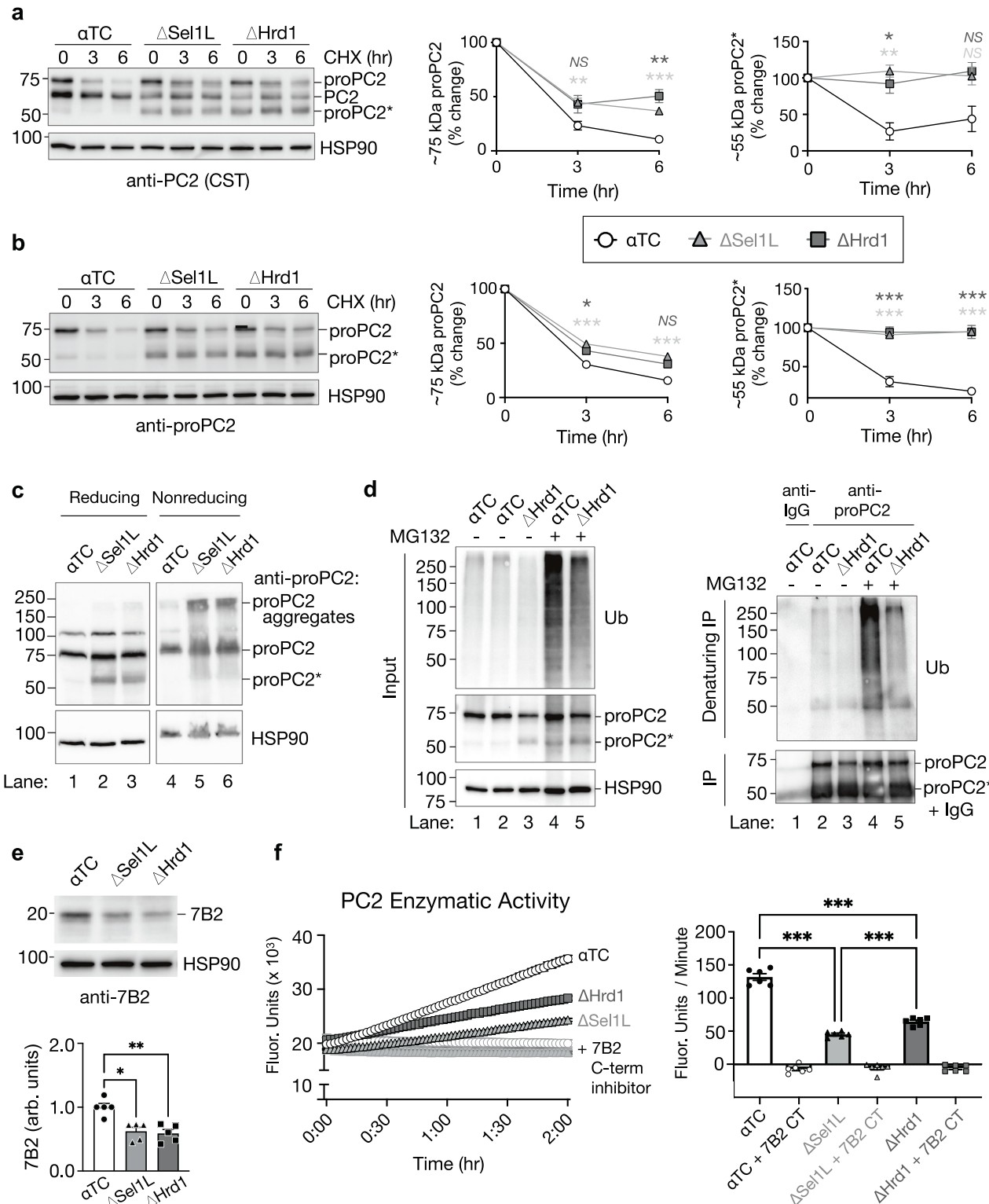

nascent proPC2 in the ER. Indeed, ERAD deficiency causes accumulation of aberrantly cleaved proPC2* and aggregation of proPC2-derived proteins, leading to reduced production and activity of mature PC2, and consequently, impaired proglucagon processing in α cells (Fig. 7). This study reveals a novel regulatory mechanism for control of PC2 function in neuroendocrine cells.

ER proteostasis is critical for optimal protein synthesis and secretion, a primary function of islet endocrine cells. Failure of islet β

cells to adapt to ER dysfunction during metabolic stress has been implicated as a key factor in diabetes pathogenesis[20], but the relative role of ER homeostasis in α cell function had not been rigorously studied. Here, we found that ER volume is increased in the expanded α cell population in *Gcg*[-/-] mice, in which the complete absence of hepatic glucagon signaling leads to hyperaminoacidemia, providing a strong stimulus for α cell proliferation[35,36]. ER volume expansion has been observed in β cells from mice and humans with diabetes[45,46] as an

**Fig. 6 | SEL1L-HRD1 ERAD degrades misfolded proPC2 in α cells to promote functional maturation of prohormone convertase 2. a, b** Western blot of αTC cells following 0, 3, and 6 h of treatment with cycloheximide (CHX) to inhibit new protein synthesis, labeled with antibodies targeting the catalytic domain of PC2 (CST, **a**) or proPC2 (**b**). Quantification of the relative expression of each (pro)PC2 form over time is shown at right (data points are averages of n = 2-4 biologic replicates from two independent experiments, shown as mean ± SEM). **c** Western blot of αTC cell lysates prepared under reducing and nonreducing conditions, labeled with antibodies to proPC2. Note that images from reducing and nonreducing conditions are from the same blot, cropped for clarity. Experiment was performed twice with separate biologic samples, with similar results. **d** Western blot analysis of protein ubiquitination before (Input, left) and after (Denaturing IP, right) immunoprecipitation with antibodies to proPC2 or IgG control in αTC and ΔHrd1 cells. Experiment was performed three times with separate biologic samples, with similar results. **e** Western blot and quantification of the proPC2 chaperone protein

7B2 in αTC, ΔSel1L, and ΔHrd1 cells. Each data point represents a single biological replicate, shown as mean ± SEM, expressed as arbitrary units (arb. units). Note that molecular weight markers are listed in kDa in (**a**–**e**). **f** PC2 enzymatic activity of cell lysates as measured by a substrate-specific aminomethylcoumarin assay, performed with and without the addition of exogenous 7B2 C-terminal (CT) peptide as a specific PC2 inhibitor. In the time course graph, each data point represents the mean ± SEM of three biological replicates in each of two independent experiments. In the bar graph, each data point represents the rate of change in fluorescence intensity for individual replicates in the time course graph, shown as mean ± SEM. For each graph in (**a**, **b**, **e**, and **f**) ΔSel1L samples are represented by light gray/triangles, ΔHrd1 represented by dark gray/squares, and αTC controls represented by white/circles. *P < 0.05; **P < 0.01; ***P < 0.001; NS, not significant (P > 0.05); two-way ANOVA with Tukey post-test was performed in **a**, **b** and one-way ANOVA with Tukey post-test was performed in **e**, **f**. Specific P values and n numbers for independent experiments are available in the source data provided as a Source Data file.

adaptation to the demand for increased insulin production[19]. As glucagon, the primary α cell peptide product, is not produced in *Gcg*[-/-] mice, it is possible that the ER expansion in this model is not solely triggered by a demand for increased protein synthesis but is instead attributable to the ER's role in supplying lipids for cell membrane expansion and growth[47]. Nevertheless, we presume that ER volume expansion would also be found in normal α cells exposed to genetic or pharmacologic glucagon receptor inhibition, and that these α cells would be predisposed to even higher demands on the ER, as (pro)glucagon production is increased in *Gcgr*[-/-] mouse models[48] and in humans with *GCGR* mutations[49] or treated with GCGR inhibitors[50].

Our understanding of the mechanisms promoting α cell ER homeostasis or ER stress remains limited. Human α cells were more resistant to inflammatory stressors that cause β cell death, despite induction of similar ER stress markers like *Ddit3* (CHOP) and *Xbp1s*[51]. Similarly, iPSC-derived α cells were more resilient to ER stress-induced apoptosis in an in vitro model of type 1 diabetes[52]. A recent analysis of single cell RNAseq data from human islets demonstrated increased activation of ER stress-related pathways in α cells from patients with type 1, but not type 2, diabetes[53]. Mouse models interrogating ER function in α cells have focused on the unfolded protein response (UPR), demonstrating distinct effects of each branch on α cell function. Germline deletion of PERK, the defective eIF2α kinase in the diabetes-causing Wolcott-Rallison syndrome, resulted in early loss of both β and α cell mass[54]. In contrast, deletion of the IRE1α effector *Xbp1* in α cells led to ER dilation and glucose intolerance in older adult mice, attributed to α cell insulin resistance causing impaired suppression of glucagon secretion[55]. The phenotypes observed in mouse models with defects in the α cell UPR are quite distinct from that of ERAD deficiency, as we have reported in other cell types – but these systems likely work in cooperation, and the ability to use compensatory mechanism(s) will determine the cell's ultimate response to a given stress[20,23,56]. For example, it remains to be determined whether the modest activation of UPR signaling in ERAD-deficient α cells, particularly through the ERAD substrate and UPR sensor IRE1α, plays any compensatory role (positive or negative) in responding to the ER defects in this model.

Islet α cells are inherently challenging to study, not only for the proportionally low numbers in murine islets but also because available genetic mouse models unavoidably target manipulation of gene expression in all cell types that make proglucagon, including enteroendocrine L-cells and neurons of the solitary tract nucleus (NTS). Despite this caveat, we were unable to detect evidence that Cre-loxP recombination occurred in L-cells, or that (if it did occur to some degree) it was of any functional consequence. Notably, a previous study also showed that serum and gut GLP-1 levels were unchanged following inactivation of the mTOR-associated protein Raptor in

proglucagon-expressing cells, in a mouse model that dramatically reduced α cell survival[57] using the same Cre line[37] as this study. Similarly, in a *Gcg*-Cre model targeting profound α cell depletion induced by diphtheria toxin, reporter gene expression was observed in much lower numbers of L-cells compared with islet α cells[58]. Together, these data suggest that Cre-mediated recombination targeted to proglucagon-expressing cells is less penetrant in L-cells compared to α cells (possibly from different levels of Cre expression caused by subtle promoter differences); alternatively, perhaps the short lifespan of L-cells lowers their dependence on a functional ERAD system. It is also unlikely that this model has impactful ERAD dysfunction in neurons given the lack of body weight changes or systemic glucose

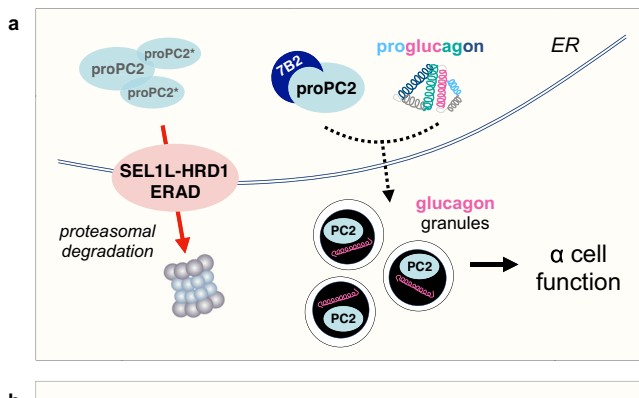

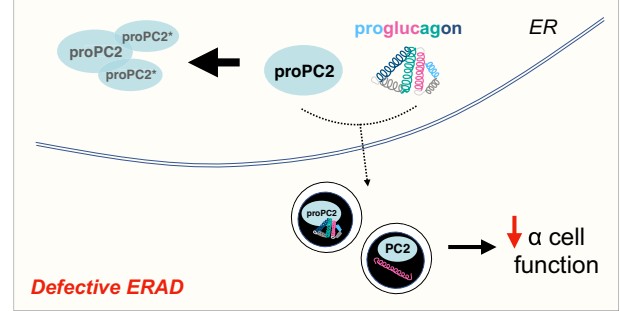

**Fig. 7 | Working model. a** SEL1L-HRD1 ERAD is a critical ER protein quality control pathway in α cells, targeting misfolded and/or aggregated pro-prohormone convertase 2 (proPC2) and aberrantly cleaved proPC2* molecules for proteasomal degradation. SEL1L-HRD1 ERAD function is essential to allow for packaging of properly folded proPC2 and proglucagon into secretory granules, where proPC2 is then activated into the functional PC2 enzyme that cleaves proglucagon into glucagon - thus promoting α cell function. **b** Defective SEL1L-HRD1 ERAD function in α cells leads to aggregation of proPC2 and aberrantly cleaved proPC2*, limiting the packaging of activation-competent proPC2 into secretory granules and thus limiting the production of mature glucagon.

dysregulation as observed with manipulation of proglucagon expression in the CNS[59] and the absence of neurologic dysfunction that we have observed in models inducing ERAD dysfunction in the brain[27,60]. Given that the mature glucagon peptide is derived nearly exclusively from islet α cells (excluding observations from pancreatectomized mammals), and given our primary finding of ERAD deficiency causing a reduction in pancreatic and circulating glucagon levels, we believe that any impact of Cre-mediated recombination in L-cells or neurons in this model is minimal.

Our data show that the effect of ERAD on proPC2 maturation is direct and uncoupled from ER stress-related protein misfolding, as chemical ER stressors failed to cause the accumulation of proPC2 or the novel truncated proPC2* protein. Instead, proPC2 and proPC2* are SEL1L-HRD1 ERAD substrates that are susceptible to formation of high molecular weight protein aggregates via aberrant disulfide bonds in the absence of ERAD function. The fact that we could detect proPC2* in control α cells after addition of MG132 and in isolated islets from $Sel1L^{fl/fl}$ mice suggests that proPC2* is generated in normal α cells but is quickly degraded by the proteasome, through SEL1L-HRD1 ERAD. Based on the estimated molecular weight of proPC2*, we predict that cleavage of proPC2 within the P domain leads to the generation of the truncated proPC2* protein; however, the specific mechanism producing proPC2* in the ER remains unclear. The neuroendocrine protein 7B2 is necessary for formation of an activation-competent proPC2 complex that can exit the ER, and in its absence proPC2 is susceptible to misfolding and aggregation in CHO cells[41]. However, previous 7B2 structure-function and knockout analyses did not show evidence of proPC2* generation[41,61,62]. Our siRNA and 7B2 overexpression experiments demonstrated that excess 7B2 alone was insufficient to rescue the accumulation of proPC2(*) in ERAD-deficient α cells, which appears to be a rapid consequence of ERAD dysfunction. We speculate that the reduction of available 7B2 cofactor in α cells with chronic ERAD deficiency exacerbates proPC2 aggregation within the ER, possibly by limiting export of nascent proPC2[41]. Future studies will be required to clearly define the role of ERAD in the regulation of 7B2 expression and its functional consequences, as well as identifying the specific mechanism by which the aggregation-prone proPC2* peptide is generated.

Our finding of a new regulator of proPC2 maturation has important therapeutic implications for diabetes and other metabolic diseases, as PC2 acts on a variety of other prohormones, including prosomatostatin, proinsulin, and proopiomelanocortin (POMC). Polymorphisms in the *PCSK2* gene encoding PC2 have been associated with abnormalities in glucose metabolism and risk of type 2 diabetes in specific groups, including African American[63], Chinese[64,65], and Old Order Amish[66] populations, though the degree of PC2 dysfunction and impact on each of its substrates is less clear. By targeting components of the ERAD pathway, such as SEL1L or HRD1, it may be possible to enhance or inhibit the degradation of proPC2, thereby modulating the levels of mature PC2 and ultimately controlling production of target hormones. Importantly, diminished ERAD activity in our model decreased glucagon production, but not the fraction of glucagon secreted upon stimulation. These data suggest that physiological release of glucagon is proportional to PC2-mediated glucagon production and storage. In α cells, strategies to enhance the function of the ERAD pathway or to stabilize proPC2 in the ER might increase the production of active PC2, leading to higher glucagon levels and improved glucose mobilization during hypoglycemic events. Alternatively, differential processing of proglucagon into glucagon-like peptide 1 (GLP-1) by α cells is a potential mechanism to promote β cell function[67,68], but whether PC2 contributes to GLP-1 production is not clear[69,70]. Hence, understanding the precise regulatory mechanisms governing proPC2 maturation and function opens new avenues for therapeutic interventions aimed at modulating hormone levels in various metabolic conditions.

Although the current work demonstrates a primary role for ERAD in regulating proPC2 maturation in α cells, our data hint at other long-term effects of ER dysfunction in α cell biology. Our CRISPR-generated ERAD-deficient α cell lines unexpectedly showed lower *Pcsk2* and *Pcsk1* gene expression, which was not observed following short-term inactivation of ERAD using an siRNA approach. These data suggest additional long-term effects on α cell identity that mimic the "dedifferentiation" we observed in ERAD-deficient β cells[30]. Similarly, we observed decreased α cell mass in older $Sel1L^{ΔGcg}$ mice, which may reflect a progressive decline in α cell number, though it is unclear if the immunofluorescence-based measurement underestimated the size of the glucagon-deficient α cell population. It remains challenging to correlate α cell mass with systemic glucagon physiology, as it has been previously shown that near-total ablation of α cells induced by diphtheria toxin dramatically reduced pancreatic glucagon content but had a lesser impact on systemic glucagon levels[71]. This suggests a robust capacity for normal α cells (e.g., those that escaped Cre-mediated recombination in our model) to maintain physiologic levels of glucagon secretion. As our female mice started with nearly double the glucagon content of male mice, similar to findings from other groups[72–74], we hypothesize that females have a greater capacity to compensate for α cell dysfunction at younger ages, but (like the males) eventually lose the ability to produce enough glucagon to mount a normal secretory response to hypoglycemia. Thus, there are several potential contributors to the glucagon production defect in $Sel1L^{ΔGcg}$ mice beyond the observed misprocessing of proPC2. The specific mechanisms underlying these additional effects remain to be explored.

In summary, we have uncovered a novel function of ERAD in the regulation of proPC2 maturation, which holds significant therapeutic potential. By modulating this system, it may be possible to develop new treatments for diabetes and other metabolic disorders. With the recent identification of human disease-causing variants of SEL1L and HRD1[27,75,76], further research into the ERAD pathway and its interactions with substrates such as proPC2 will be crucial in translating these findings into clinical therapies, ultimately improving outcomes for patients with a range of conditions.

## Methods

### Study approval
All animal procedures were approved by and done in accordance with the IACUC at the University of Michigan Medical School (PRO00008989/PRO00010658/PRO00011495).

### Mice
$Sel1L^{fl/fl}$ mice[24] on the C57BL/6 J background were crossed with B6;129S-$Gcg^{tm1.1(icre)Gkg}$/J ($Gcg^{iCre}$) mice[37] to generate mice with SEL1L deletion in proglucagon-expressing cells (denoted $Sel1L^{ΔGcg}$ herein), with Cre-negative ($Sel1L^{fl/fl}$) or heterozygous ($Sel1L^{ΔGcg/+}$) littermates used as controls. $Gcg^{iCre}$ mice were also crossed with B6.129×1-$Gt(ROSA)$$26Sor^{tm1(EYFP)Cos}$/J reporter mice (Jackson Laboratory 006148). Glucagon-STOP-flox ($Gcg^{-/-}$) mice, containing GFP and a poly(A) "stop" signal between exons 2 and 3 of the *Gcg* gene, were previously described[34]. Mice were housed in an ambient temperature room with a 12 hr light cycle and fed a normal-chow diet (13% fat, 57% carbohydrate, and 30% protein, LabDiet 5L0D). Males and females were used equally in experiments, except where sexually dimorphic results were observed and reported.

### Human islets
Isolated islets from cadaveric donors were obtained from the Human Islet Resource Center at the University of Pennsylvania through the Human Pancreas Analysis Program[77,78] and Integrated Islet Distribution Program (IIDP)[79], following the guidelines of the Clinical Islet Transplantation Consortium protocol (https://www.isletstudy.org). Briefly, the pancreas was digested following an intraductal injection of

collagenase & neutral protease in Hanks' balanced salt solution and then purified on continuous density gradients (Cellgro/Mediatech) using a COBE 2991 centrifuge. Islets were cultured in CIT culture media and maintained in a humidified 5% CO2 incubator. The islets were distributed per the Integrated Islet Distribution Program (IIDP) protocols. After receipt, islets were hand-picked and cultured overnight prior to embedding in low melting point agarose[80] before fixation in 1% paraformaldehyde and mounting in O.C.T. medium in preparation for cryosectioning and immunohistochemistry as previously described[81,82]. Gift of Life, along with other organ procurement organizations, obtained consent from the deceased donors' families for the use of organs in research. All procedures complied with the University of Pennsylvania IRB, Gift of Life leadership team, and UNOS regulations and standards. Donor information is presented in Supplementary Table 2. Only male donors were available for this study.

### Glucose tolerance tests and in vivo glucagon and GLP-1 secretion

Mice were fasted for 4-6 hr prior to each experiment. Blood was sampled via tail nick. For glucose tolerance testing, blood glucose was measured at baseline prior to intraperitoneal or oral gavage-mediated administration of D-glucose at 1.5 g/kg body weight. Blood glucose was measured 15, 30, 60, and 90 min after glucose administration using a OneTouch Ultra glucometer. For assessment of glucagon secretion, blood was collected in heparinized tubes before and 30 minutes following intraperitoneal injection of 0.75 units/kg insulin (Novolin R, Fisher Scientific NC0769896). Serum was collected by centrifugation, immediately frozen at -80 °C, and thawed prior to measurement by ELISA (Mercodia 10-1281-01). For assessment of total serum GLP-1 levels, mice were given an oral gavage of 3 g/kg D-glucose ten minutes before undergoing isoflurane anesthesia and cardiac puncture. DPPIV inhibitor (Millipore Sigma DPP4) was added to whole blood prior to centrifugation, and serum was frozen at -80 °C prior to measurement by U-PLEX biomarker assay (Meso Scale Diagnostics K1525UK). Note that negligible levels of glucagon and GLP-1 were detected in pancreas extracts or serum from $Gcg^{-/-}$ mice using these assays (Supplementary Fig. 1d).

### Pancreatic and gut hormone content

Immediately following sacrifice by cervical dislocation, mouse pancreata were dissected, trimmed of fat, weighed, placed in 3 ml of acid alcohol (1.5% hydrochloric acid in 75% [v/v] ethanol in water) and homogenized for 30 sec. The suspension was rotated for 48 hr at 4 °C, centrifuged at $1540 \times g$ for 30 min at 4 °C, then the supernatant stored at -80 °C until further analysis. For GLP-1 measurement in gut, the most distal 3 cm of colon was dissected, and gut epithelium was scraped with a coverslip and placed in either acid alcohol or T-PER extraction reagent (Thermo Scientific 78510) with DPPIV inhibitor as previously described[34] for hormone extraction. Glucagon and/or GLP-1 content of diluted supernatant was measured by ELISA, as above. Pancreatic insulin content was measured by ELISA (Crystal Chem 90080).

### Histology and islet morphometry

Pancreata were isolated, fixed in 10% neutral buffered formalin (VWR 95042-908) or 4% paraformaldehyde (Electron Microscopy Sciences 15710) for 1.5–16 h at 4 °C. Samples for paraffin embedding and sectioning were dehydrated in 70% ethanol and processed by the University of Michigan Comprehensive Cancer Center Tissue and Molecular Pathology Shared Resource. Alternatively, some pancreata were processed for cryosectioning as previously described[83]. Measurements of α- and β-cell mass were performed by the University of Michigan Islet Core. Briefly, five sections per pancreas, each at least 150 μm apart, were labeled with antibodies to insulin (Agilent, Santa Clara, CA) and glucagon (Immunostar, Hudson, WI), then imaged by tiling with a computer-controlled AZ-100 microscope with motorized stage using the NIS Elements AR software (Nikon, Cambridge, MA) and a

Coolsnap EZ Digital Camera (Photometrics, Tucson, AZ). After acquisition, α- and β-cell areas were determined by intensity thresholding with manual curation using FIJI 2.14.0. Total α- and β-cell mass were then calculated using the following calculation: [(signal area/total pancreatic section area) x pancreas weight].

### Immunofluorescence staining

Deparaffinized tissue sections and cryosections were immunolabeled as described previously[30]. The following primary antibodies were used: SEL1L (homemade[30,82], 1:200), insulin (Bio-Rad 5330-0104 G, 1:500), glucagon (Peninsula Labs/BMA Biomedicals T-5037, 1:200), proglucagon (Cell Signaling Technology 8233S, 1:200), transthyretin (Invitrogen PA580196/PA580197, 1:500), somatostatin (Abcam ab30788, 1:200), BiP (Abcam 21685, 1:100), OS9 (Abcam ab109510, 1:100), GFP (Abcam 13970, 1:2000), PC2 (Cell Signaling Technology 14013 s, 1:800), proPC2 (homemade[14], 1:200), GLP-1 (Peninsula Labs/BMA Biomedicals T-4363, 1:200), E-cadherin (BD Biosciences 610181, 1:100), PC1/3 (homemade RS20[84–86], 1:200). Details on antigens for key antibodies can be found in Supplementary Table 1. Sections were treated with species-specific secondary antibodies conjugated to Alexa Fluor dyes (Jackson ImmunoResearch; 1:500) followed by mounting with VectaShield (Vector Laboratories, H-1500) or Prolong Gold (Invitrogen P36931) mountant with DAPI. Immunolabeled pancreatic sections were imaged using a Leica DM6000 through the University of Michigan Vision Research Center Morphology and Imaging Core, and Nikon A1 or Leica Stellaris confocal microscopes at the University of Michigan Morphology and Image Analysis Core. Images were prepared for publication using ImageJ software (NIH, version 2.14.0) and figure layouts created in Numbers (MacOS, version 14.4). Immunofluorescence quantification was performed using Imaris software (v10.0, Oxford Instruments).

### Proliferation and TUNEL assays

Paraffin embedded pancreas sections were immunolabeled with Ki67 (Abcam 15580; 1:100) and glucagon to detect proliferating α cells. TUNEL assay was performed using the In-Situ Cell Death Detection Kit (Roche 11684795910), with glucagon co-labeling by immunofluorescence. Images were acquired using a Leica DM6000 microscope for counting 15-100 islets per mouse to capture a total of 2000–4000 α cells ($n = 3–5$ per genotype). Cell proliferation and apoptosis rates were calculated as a percentage of Ki67- or TUNEL-positive cells, respectively, per total number of glucagon-positive cells.

### Transmission electron microscopy

Whole pancreas was rapidly dissected, placed in fixative (2.5% glutaraldehyde and 4% paraformaldehyde in 0.1 M Na-cacodylate buffer), minced into ~1–2 mm cubes, and fixed overnight at 4 °C followed by washes in buffer. Thereafter, samples were submitted to the University of Michigan Microscopy Core for post-fixation in 1% osmium tetroxide, embedding in resin, sectioning, and mounting on grids. Imaging was performed using a JEOL JEM 1400 PLUS microscope with an AMT NanoSprint 12 scientific grade CMOS camera.

### Islet isolation and secretion

Pancreatic islets were isolated from mice as previously described[87]. Briefly, mice were sacrificed by cervical dislocation and pancreata perfused by intraductal injection of 2.5 mL collagenase (Liberase TL [Roche 5401020001] in serum-free RPMI 1640 [Gibco 11875-085]). Pancreata were incubated at 37 °C for ~12 min with an additional 1 mL of digestion solution. Digestion was stopped by adding cold HBSS with 10% FBS. Islets were washed and hand-picked under a dissecting microscope, then allowed to recover overnight in RPMI medium in a humidified incubator (95% air, 5% CO$_2$) at 37 °C prior to further analysis. For static secretion experiments, 10 size-matched purified islets were incubated in 200 μl KRBB (Krebs-Ringer Solution, HEPES-

buffered, Thermo-Fisher Scientific J67795.K2) containing D-glucose +/- L-arginine hydrochloride (MilliporeSigma A5131) and media collected at the indicated times. Following the experiments, islets were lysed (50 mM Tris HCl pH 7.4, 1 mM EDTA, protease inhibitor [MilliporeSigma P8340], phosphatase inhibitor [MilliporeSigma P5726]), total protein concentrations determined by Pierce BCA assay (Thermo Fisher Scientific), and media and lysate glucagon levels measured by ELISA as above.

## Cell culture and Western blotting

Control αTC(1-6) cells were a kind gift from Ernesto Bernal-Mizrachi and also independently obtained from ATCC (CRL-2934). Cells were cultured in αTC complete medium per manufacturer recommendations. CRISPR-targeting sgRNA oligonucleotides designed against mouse *Sel1l*, *Syvn1*, or *Pcsk2* (sequences listed in Supplementary Table 3) were inserted into the lentiCRISPR, version 2 vector (Addgene, 52961) prior to transfection in αTC (or HEK293T) cells and puromycin selection of modified cells as previously described[76]. Cells transfected with the empty lentiCRISPR vector were used as a control. In some experiments, cells were treated with cycloheximide (50 μg/ml; Sigma-Aldrich C7698), endoglycosidase H (New England Biolabs P0702S), or PNGase F (New England Biolabs P0704S) per manufacturer's instructions. In siRNA treatment experiments, control αTC cells were transfected with 30 pmol of siRNA targeting *Sel1l*, *Syvn1*, or a Silencer Select Negative Control (Invitrogen s73510, s92312 and 4390844) using RNAiMAX (Lipofectamine RNAiMAX Transfection Reagent, #13778075). In the 7B2 overexpression experiments, CRISPR-treated αTC cells were seeded in 6-well plates and cultured until reaching 70-90% confluency. For each well, two 125 μl aliquots of Opti-MEM (Gibco, 31985062) were prepared. One aliquot was used to dilute 3.75 μl of Lipofectamine 3000 (Invitrogen, L3000015), while the other was used to dilute 5 μg of the 7B2 plasmid DNA (From Iris Lindberg Lab) combined with 10 μl of P3000 reagent (Invitrogen, L3000015). The DNA-lipid complexes were then mixed and then added to the cells. Upon experiment completion (typically 48 h after siRNA or 7B2 plasmid), cells were lysed in NP-40 lysis buffer (50 mM Tris HCl at pH 7.5, 150 mM NaCl, 1% NP-40, 1 mM EDTA) with protease inhibitor (MilliporeSigma P8340), DTT (MilliporeSigma, 1 mM), and phosphatase inhibitor (MilliporeSigma P5726) cocktail. Lysates were incubated on ice for 30 min and centrifuged at $16,000 \times g$ for 10 min. Supernatants were collected and protein concentrations determined using Bradford assay (Bio-Rad). For reducing SDS-PAGE analysis, proteins were heat-denatured at 65-90 °C for 10 min in 5x SDS sample buffer (250 mM Tris-HCl pH 6.8, 10% sodium dodecyl sulfate, 0.05% bromophenol blue, 50% glycerol, and 1.44 M β-mercaptoethanol). For non-reducing conditions, lysates were prepared at 37 °C with 5x non-denaturing sample buffer (250 mM Tris-HCl pH 6.8, 1% sodium dodecyl sulfate, 0.05% bromophenol blue, 50% glycerol) for 60 min[27]. Samples were resolved by gel electrophoresis, then transferred to PVDF membranes (Bio-Rad). Some membranes were fixed with PBS containing 4% paraformaldehyde and 0.01% glutaraldehyde for 30 min[88]. Membranes were blocked with 5% milk in Tris-buffered saline Tween-20 (TBST) for 30 min, then incubated overnight at 4 °C with antibodies prepared in 2% bovine serum albumin (BSA) in TBST. The following primary antibodies were used: anti-SEL1L (homemade[30], 1:10,000), anti-HRD1 (Proteintech 13473-1, 1:2,000), anti-OS9 (Abcam ab109510, 1:5,000), anti-BiP (Abcam ab21685, 1:5,000), anti-IRE1α (Cell Signaling Technology [CST] 3294, 1:2,000), anti-PERK (CST 3192, 1:1000), anti-eIF2α (CST 9722, 1:1000), anti-phospho-eIF2α (CST 9721, 1:1000), anti-PC2 (CST 14013 s, 1:500), anti-PC2 C-terminus (homemade[39], 1:200), anti-proPC2 (homemade[14], 1:200), anti-7B2 (homemade[61], 1:200), anti-PC1/3 (Cell Signaling 18030S, 1:200), anti-glucagon (ABclonal A22702, 1:5000), anti-ubiquitin (Santa Cruz, sc-8017), and anti-HSP90 (Abcam ab13492; 1:2000). Details on antigens for key antibodies can be found in Supplementary Table 1. Membranes were washed and then incubated with goat anti-rabbit or anti-mouse IgG-HRP (Bio-Rad 1706515 and 1706516, 1:3000-1:5000) in 5% milk in TBST for 1 hr at room temperature, washed, and labeled with Clarity ECL substrate (Bio-Rad). Blots were imaged with a Bio-Rad ChemiDoc and protein band intensity was quantified using Image Lab software (Bio-Rad, version 6.1).

## Immunoprecipitation and ubiquitination assay

Assay was performed as previously described[75]. Briefly, control and ΔHrd1 αTC cells were treated with 10 μM MG132 (Enzo, BML-PI102) for 2 hr. Whole cell lysates were prepared in non-reducing lysis buffer as above. The supernatant was denatured with 1% SDS and 5 mM DTT at 95 °C for 10 min, then diluted 1:10 with NP-40 lysis buffer and mixed with control IgG antibody (Cell Signaling Technology 2729) or anti-proPC2 antibody (as above). This mixture was incubated with 15 μl anti-protein A agarose (Millipore Sigma, 11719408001) overnight at 4 °C with gentle rocking. Agarose beads were washed three times with NP-40 lysis buffer and eluted in the SDS sample buffer at 95 °C for 5 min, followed by SDS-PAGE and Western blotting as above.

## PC2 enzyme assay

A substrate-specific PC2 assay was performed as previously described[44]. Control, ΔSel1L, and ΔHrd1 αTC cells were washed and sonicated in ice-cold lysis buffer (0.1 M sodium acetate, pH 5.5, 1% Triton X-100, 50 μM trans-epoxysuccinyl-L-leucylamido[4-guanidino] butane [E-64], 50 uM pepstatin, 1 mM phenylmethylsulfonyl fluoride [PMSF]) and centrifuged at $10,000 \times g$ at 4 °C for 10 minutes. The supernatant was transferred to clean microcentrifuge tubes and protein concentrations determined using a Bradford assay (Bio-Rad). Samples of equal protein concentration (10 μg) were added to a polypropylene microtiter plate with 40 μl reaction buffer mix (2 mM $CaCl_2$, 0.1 M sodium acetate buffer [pH 5.0], 0.1% Triton X-100, 5.6 mM Nα-tosyl-L-lysine chloromethyl ketone hydrochloride [TLCK], 6 mM N-p-tosyl-L-phenylalanine chloromethyl ketone [TPCK], 2 mM E-64, 2 mM pepstatin). Each assay was performed in duplicate, both with and without the synthetic 7B2 C-terminal peptide (7B2-CT). The reaction was preincubated for 15 min at 37 °C, followed by addition of 5 μl of substrate, pGlu-Arg-Thr-Lys-Arg-MCA (RTKR-MCA, final concentration 0.2 mM in dimethyl sulfoxide [DMSO]). The reaction mixture was briefly mixed and the fluorescence of the liberated aminomethyl-coumarin (AMC; excitation 380 nm, emission 460 nm) was documented every two minutes using a fluorometer.

## RNA extraction, microarray, cDNA synthesis and qPCR analysis

RNA was isolated by phenol extraction and concentration determined using a NanoDrop 2000 UV-Vis Spectrophotometer. RT-PCR for assessment of *Xbp1* mRNA splicing and qPCR analyses were performed as previously described[89]. All PCR data were normalized to the ribosomal *L32* or *RplpO* gene expression level. All qPCR primers were obtained from Integrated DNA Technologies (IDT) and sequences are shown in Supplementary Table 4.

## Protein structure analysis

Protein structures were predicted using the AlphaFold 3 server, located at https://alphafoldserver.com/[90]. Images were rendered with PyMOL (version 2.3.2).

## Statistical analysis, reproducibility, and figure generation

Results are expressed as mean ± SEM unless otherwise stated. All experiments were repeated at least twice and performed with several independent biological samples as noted in the figure legend and/or Source Data file, with representative data shown. Statistical analyses and data graph generation were performed in GraphPad Prism (GraphPad Software Inc., version 10.6.1). Comparisons between two groups were made by unpaired two-tailed Student's t test. Comparisons between multiple groups were made by one- or two-way ANOVA

followed by Šidák or Tukey post-test, as noted. *P* values less than 0.05 were considered statistically significant. The model in Fig. 7 was generated in the MacOS Numbers program (version 14.4).

### Reporting summary

Further information on research design is available in the Nature Portfolio Reporting Summary linked to this article.

## Data availability

The materials and reagents used are either commercially available or are available upon request. All data and materials for the manuscript are described in Methods. Values for all datapoints in graphs and full blots are reported in the Source Data file. Source data are provided with this paper.

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

## Acknowledgements

We thank Dr. Ernesto Bernal-Mizrachi for sharing αTC(1-6) cells; Dr. Yewei Ji, Dr. Leena Haataja, Jianing Zhang, Elise Corden, and Steve Lentz for experimental support; and all other members of the Qi and Arvan laboratories for comments and technical assistance. This work was completed with the assistance of NIH-supported core facilities, including the Microscopy, Imaging and Cellular Physiology Core (MICPC) and Islet Isolation Laboratory supported by the National Institute of Diabetes, Digestive, and Kidney (NIDDK)-funded Michigan Diabetes Research Center (P30DK020572 and Shared Instrument Grant S10OD28612-01-A1), the University of Michigan Vision Research Center Morphology and Imaging Core supported by the National Eye Institute (P30EY007003), the Tissue and Molecular Pathology Shared Resource supported by the National Cancer Institute (P30CA04659229), and the University of Michigan Biomedical Research Core Facilities Microscopy Core. This manuscript used human islets acquired from the University of Pennsylvania Islet Transplant Center in collaboration with the Human Pancreas Analysis Program (HPAP-RRID:SCR_016202), a Human Islet Research Network (RRID:SCR_014393) consortium (UC4DK112217), and the Integrated Islet Distribution Program (IIDP) (RRID:SCR_014387) through City of Hope (UC4DK098085), supported by Beckman Research Center grant 10028044 (to A.N.). We thank the human islet donors and their families for their generous contribution. This work was supported by R24DK110973 (to P.A.), R01DK11174 and 2-SRA-2018-539-A-B (to P.A. and L.Q.), R01DK121995 (to D.A.S), R01DK137794 and R35GM130292 (to L.Q.), and 5T32DK007245 and K08DK129719 (to R.B.R.).

## Author contributions

W.Z., L.P., X.C., A.C.R., R.R., B.P., X.W., L.L.L., M.T., H.H., B.G., and R.B.R. performed experiments that were designed by W.Z., L.P., L.Q., and R.B.R.; N.S. made the initial crosses of knockout mice and provided experimental support and helpful discussion; C.L., A.N., P.A., D.S., and I.L. provided key reagents and helpful discussion; L.Q. and R.B.R. supervised the project and wrote the manuscript. All other authors edited and approved the manuscript.

## Competing interests

The authors declare no competing interests.
