## [Transparent Peer Review file · Nature Communications]

SEL1L-HRD1 ER-Associated Degradation Facilitates Prohormone Convertase 2 Maturation and Glucagon Production in Islet α Cells

Corresponding Author: Dr Rachel Reinert

Version 0:

Reviewer comments:

Reviewer #1

(Remarks to the Author)

This study investigates a novel role for the SEL1L-HRD1 ER-associated degradation (ERAD) system in regulating prohormone convertase 2 (PC2), which is required for processing proglucagon in pancreatic alpha cells. The authors provide evidence that SEL1L-HRD1 ERAD is essential for clearing excess proPC2. They also demonstrate that dysfunctional SEL1L-HRD1 ERAD leads to accumulation of a C-terminally truncated proPC2 form, which the authors speculate is an aberrantly cleaved/misfolded PC2 that is not properly cleared by ERAD. The study observes a progressive decline in stimulated glucagon secretion and a reduction in pancreatic glucagon content in mice lacking Sel1L, likely linked to PC2 dysfunction. The experiments are well-performed and appropriately controlled. The presence and rapid clearance of the truncated proPC2 form is an interesting observation, as well as the finding that ERAD mediated clearance of proPC2 is seemingly decoupled from a general ER stress response. These findings may provide mechanistic insight into the consequences of human disease variants of ERAD components, or potentially an avenue for controlling glucagon output in diabetes by modulating ERAD / PC2. The study's findings are novel and of potential importance albeit perhaps not too surprising given that SEL1L-HRD1 ERAD is a highly conserved system and proPC2 is already known to be prone to aggregation and accumulation. There are a few issues that should be addressed to improve the manuscript and strengthen its conclusions.

Major Concerns:

1. Some of the important conclusions seem to be dependent on immunostaining data and/or use of antibodies with specificity that is not known or at least described. Complementary biochemical assays for identifying proglucagon forms such as Western blotting, and better characterization of antibody specificity should be provided, as per some specific examples below:
 - (a) total PC2 immunoreactivity is stated to be increased in Fig 4D in alpha cell Sel1L KO alpha cells, but this does not seem to match with the aTC KO western data in Fig 5 where proPC2 is increased but total PC2 seems similar. It would be helpful to perform westerns as well on the alpha cell Sel1L KO primary islets (contribution of beta cells to PC2 should be small) to assess proPC2 processing and total PC2 content, and quantification of total PC2 from westerns in Figure 5.
 - (b) The proglucagon and glucagon antibodies used in immunostaining are not detailed or their cross-reactivity with other antibodies, and the field is known for cross-reactivity of these antibodies with different molecular forms of proglucagon/glucagon derived peptides. Western blot of islet lysates from KO and control mice and mutated aTC cells to better understand the molecular forms of proglucagon peptides would be informative.
2. PC2-glucagon link: The data indicating that proPC2 is an ERAD substrate is convincing, but the evidence that accumulation of proPC2 and reduction in PC2 activity due to ERAD disruption is the direct cause of loss of glucagon in these models would benefit from additional supporting data or at least explanation, for example:
 - (a) If ERAD knockout is causing significant PC2 activity loss, an increased accumulation of proglucagon should be evident, and demonstrated by western blot.
 - (b) The authors could strengthen their conclusions by showing that loss of PC2 activity with PC2 deletion and/or 7B2 inhibitor alters proglucagon production / processing.
 - (c) As detailed in 1 above, demonstration by western in their models that proglucagon processing is impaired, as immunostaining and ELISAs with available antibodies have unknown or questionable specificity.
 - (d) The decrease in total pancreatic glucagon and proglucagon shown in Figure 3D does not seem to account for the significant and near proportional loss of alpha cell mass in Fig S3D. Could the loss of glucagon be simply related to loss of

alpha cell mass?

(e) Why does the functional glucagon defect only manifest in older Sel1L Δ Gcg mice (4 months male; 8 months female)? If a proPC2 processing defect is truly the cause of glucagon dysfunction, would the authors not expect the glucagon defect to be apparent earlier? One interpretation might be due to loss of alpha cells with decreasing with age associated with ERAD disruption. The authors should assess alpha cell population in younger mice. Further, performing cell viability and proliferation assays on the aged mice might be influenced by a survivorship bias.

(f) Measurement (ELISA) of GLP-1 in ERAD deficient alpha cells would be informative. What happens to Pcsk1 / PC1/3 in ERAD deficient alpha cells? It would be of interest if the defect was specific for PC2 and resulted in increased GLP-1 production, though this is not clear.

(g) Does Pcsk2 mRNA expression change in the mutants?

3. Concerns with some figures:

(a) Fig 1C – inclusion of insulin staining in top panel of 1C (control) would enable clearer comparison with the Gcg KO panel below.

(b) representative images in Fig 2 could be more representative: in Fig 2C it is hard to see BiP expression changes when there are so many more alpha cells in the control than in the Sel1L KOs; Fig 2D does not convincingly show the slight but significant decrease in proglucagon and glucagon levels in Sel1L knockouts reported by the quantification.

(c) The presence of a proPC2 band in the PC2 knockout condition is puzzling and would benefit from explanation.

Minor

1. The implication that SEL1 is not changed in type 2 diabetes should be softened since it is based on immunostaining from just one WT and one diabetic donor, or more donors included.

2. Please state the species of the homemade proPC2 antibody in the methods section.

3. Quantification of OS9 expression in Figure 2B in a manner similar to BiP in Figure 2C to provide would strengthen conclusion that it is upregulated.

Reviewer #2

(Remarks to the Author)

This manuscript by Zhu et al investigates the role of the SEL1L-HRD1 ERAD pathway in islet α cells, demonstrating that it is essential for the proper folding, maturation, and enzymatic activation of proPC2, a key enzyme in glucagon biogenesis. The authors provide evidence that Sel1L and Hrd1 knockout in proglucagon-expressing cells leads to aberrant proPC2 processing, accumulation of a truncated proPC2* species, decreased PC2 activity, and ultimately impaired proglucagon cleavage. These findings reveal a post-translational regulation in α cell biology and open potential therapeutic avenues in metabolic diseases.

The paper is clearly structured: the introduction lays out the need for this study, methods are suitable to test their hypothesis, the data support the main conclusions, and discussions are thorough.

Several mechanistic and contextual points should be addressed to solidify the conclusions.

Major concerns:

1. Mechanism of proPC2* generation

The accumulation of proPC2* in ERAD-deficient α cells is intriguing and raises important questions about its origin. While the proposed C-terminal cleavage is plausible, the exact mechanism by which proPC2* is generated remains unclear. Since proPC2* appears to be a potentially important intermediate with pathological relevance, it would greatly strengthen the study to provide further biochemical insight into how this fragment is formed—whether through a specific protease, misfolding, or another pathway.

2. Role of 7B2 downregulation and feedback loops

The observed downregulation of 7B2 in ERAD-deficient α cells suggests a contributing factor to impaired PC2 maturation. However, this conclusion is largely correlative. The authors are encouraged to assess whether exogenous 7B2 expression can rescue PC2 activity or reduce proPC2 aggregation. This would provide functional support for the 7B2–PC2 axis as a downstream effector of ERAD in α cells.

3. Rule out the possible off-target effects of the Gcg-Cre model in NTS

Given the use of a Gcg-Cre model, the potential for off-target recombination in L-cells and the central nervous system (e.g., NTS) must be considered carefully. While the authors reference prior studies and the absence of metabolic phenotypes, additional lineage tracing or tissue-specific expression data would help rule out confounding effects.

Recommendation: Major Revision

Version 1:

Reviewer comments:

Reviewer #1

(Remarks to the Author)

The authors have done a thorough job of responding to this reviewer's concerns. While I still have minor reservations about

the (pro)glucagon antibodies and their specificity for different molecular forms, the authors have done about as much as can be done with available resources and have described it well with new data and text.

Reviewer #2

(Remarks to the Author)

The authors have adequately addressed my concerns. I commend the authors for the high quality and scientific rigor of their work. Overall, I find the study to make a valuable contribution to the field and recommend it for publication.

RESPONSE TO REVIEWER COMMENTS

Reviewer #1 (Remarks to the Author):

This study investigates a novel role for the SEL1L-HRD1 ER-associated degradation (ERAD) system in regulating prohormone convertase 2 (PC2), which is required for processing proglucagon in pancreatic alpha cells. The authors provide evidence that SEL1L-HRD1 ERAD is essential for clearing excess proPC2. They also demonstrate that dysfunctional SEL1L-HRD1 ERAD leads to accumulation of a C-terminally truncated proPC2 form, which the authors speculate is an aberrantly cleaved/misfolded PC2 that is not properly cleared by ERAD. The study observes a progressive decline in stimulated glucagon secretion and a reduction in pancreatic glucagon content in mice lacking Sel1L, likely linked to PC2 dysfunction. The experiments are well-performed and appropriately controlled. The presence and rapid clearance of the truncated proPC2 form is an interesting observation, as well as the finding that ERAD mediated clearance of proPC2 is seemingly decoupled from a general ER stress response. These findings may provide mechanistic insight into the consequences of human disease variants of ERAD components, or potentially an avenue for controlling glucagon output in diabetes by modulating ERAD / PC2. The study's findings are novel and of potential importance albeit perhaps not too surprising given that SEL1L-HRD1 ERAD is a highly conserved system and proPC2 is already known to be prone to aggregation and accumulation. There are a few issues that should be addressed to improve the manuscript and strengthen its conclusions.

We thank the reviewer for these positive comments and for the feedback below, which has greatly improved our manuscript.

Major Concerns:

1. Some of the important conclusions seem to be dependent on immunostaining data and/or use of antibodies with specificity that is not known or at least described. Complementary biochemical assays for identifying proglucagon forms such as Western blotting, and better characterization of antibody specificity should be provided, as per some specific examples below:

We agree that a better description of antibody specificity is helpful and have added **Supplementary Table S1** to summarize the key peptide antigen targets and provide references to representative publications outlining the generation and/or use of these antibodies. We have also added clarifying text to the figure legends where necessary, outlining the specific antibodies used.

(a) total PC2 immunoreactivity is stated to be increased in Fig 4D in alpha cell Sel1L KO alpha cells, but this does not seem to match with the aTC KO western data in Fig 5 where proPC2 is increased but total PC2 seems similar. It would be helpful to perform westerns as well on the alpha cell Sel1L KO primary islets (contribution of beta cells to PC2 should be small) to assess proPC2 processing and total PC2 content, and quantification of total PC2 from westerns in Fig. 5.

We agree that this would be useful information. We have added quantification for combined isoforms of (pro)PC2 in Δ Sel1L and Δ Hrd1 α cells in **new Fig. S5a-b**, which confirms that “total” PC2 is slightly increased in Δ Sel1L cells.

In addition, we performed Western blotting on islet lysates from *Sel1L* ^{Δ Gcg} mice and littermate controls and found a similar increase in expression of both proPC2 and proPC2*, reflecting accumulation of these isoforms as also observed in Δ Sel1L α TC cells. We have **added these data to Fig. S5c** and added relevant text (pertaining to both sets of new data) to the Results on page 8.

NEW Figure S5a-c. a-b Quantification of combined proPC2 and proPC2* isoforms (**a**) and all (pro)PC2 isoforms (“Total PC2,” **b**) in α TC-1-6 cells following CRISPR-mediated deletion of Sel1L (Δ Sel1L) or Hrd1 (Δ Hrd1)

compared to vector-treated α TC controls. **c** Western blot of isolated islet lysates from *Sel1L* ^{Δ Gcg} mice and *Sel1L*^{fl/fl} controls, labeled with antibodies to proPC2. Quantification of the protein bands normalized to HSP90 is shown at right. Each data point in the graphs represents an individual replicate. * $P < 0.05$, ** $P < 0.01$, *** $P < 0.001$; NS, not significant ($P > 0.05$); unpaired two-tailed Student's t-test or one-way ANOVA with Šidák post-test.

Revised text, page 8: “The combined increase in full-length proPC2 and the proPC2 fragment led to a slight increase in the overall expression of proPC2-derived proteins in Δ Sel1L α cells (Supplementary Fig. 5a-b), reflecting the pattern observed in *Sel1L* ^{Δ Gcg} islets. We confirmed that proPC2* expression was also increased in *Sel1L* ^{Δ Gcg} islets by Western blot (Supplementary Fig. 5c).”*

(b) The proglucagon and glucagon antibodies used in immunostaining are not detailed or their cross-reactivity with other antibodies, and the field is known for cross-reactivity of these antibodies with different molecular forms of proglucagon/glucagon derived peptides. Western blot of islet lysates from KO and control mice and mutated α TC cells to better understand the molecular forms of proglucagon peptides would be informative.

The reviewer raises an excellent point; we agree that these considerations for antibodies to proglucagon-derived peptides (PDPs) need to be taken into account. The known targets for antibody generation are **now listed in Table S1**, along with references to published manuscripts using these antibodies. Although many antibodies to glucagon and GLP-1 are targeted to their mature sequences, we acknowledge a likely crossreactivity with the lesser-studied PDPs that also contain these sequences (e.g., glicentin, oxyntomodulin, and the major proglucagon fragment) as well as the full-length proglucagon sequence.

As such, we attempted to detect the specific PDPs by Western blotting using the anti-glucagon antibodies that successfully labeled alpha cells by immunofluorescence in our study (though noting that none of these commercial antibodies were previously validated for this method). We were unable to reliably detect the ~3 kDa mature glucagon by Western blotting using the antibodies from Cell Signaling Technology or BMA Biomedicals, though we did detect the full-length proglucagon band with the BMA anti-glucagon antibody (**new Fig. S1c**). Thus, we have revised all figures with immunofluorescence data to reflect the detection of “(pro)glucagon” by this technique.

NEW Figure S1c. Western blot of αTC cells with or without CRISPR-mediated inactivation of PC2 using the BMA Biomedicals T-5037 anti-glucagon antibody, revealing reliable detection of the full-length proglucagon protein but no detectable bands at the expected ~3 kDa location for mature glucagon.

We remain confident in the specificity of the anti-proglucagon antibody given its lack of immunofluorescence in *Gcg*^{-/-} (global knockout) mice (Fig. S1b). In support, this anti-proglucagon antibody has been used to label alpha cells in mice harboring a specific deletion of the exons encoding mature glucagon, as typical anti-glucagon antibodies are nonreactive in this model (Tellez et al., *Nat Metab* 2020, PMC7739959; Coate et al., *Mol Metab* 2024, PMC11570739; both **referenced in new Table S1**).

While the quantification of immunofluorescence images supports our primary claim that SEL1L-HRD1 ERAD is essential to maintain glucagon production in alpha cells, we emphasize that our strongest evidence is derived from quantitation of mature glucagon in serum and pancreas/islet extracts by ELISA. The Mercodia sandwich ELISA kit in this work uses a dual antibody system to target both ends of the mature glucagon sequence and reports low cross-reactivity with mouse glicentin (7.0%), oxyntomodulin (2.0%), and GLP-1 (not detected) (as reported by the company at <https://www.mercodia.com/products/glucagon-elisa-10-µl/>). Although there have been reports for overestimation of glucagon detection with this assay, this has mainly been observed in human patients who have undergone bariatric surgery and have elevated secretion of (gut-derived) PDPs like glicentin (Albrechtsen et al., *Scand J Clin Lab Invest* 2022, DOI: 10.1080/00365513.2021.2016943). We have confirmed that pancreas extracts from *Gcg*^{-/-} mice give negligible readings with this assay (**new Fig. S1d**; this point has been added to the Methods section on page 17).

*Revised text, page 17: “Note that negligible levels of glucagon and GLP-1 were detected in pancreas extracts or serum from *Gcg*^{-/-} mice using these assays (Supplementary Fig. 1d).”*

2. PC2-glucagon link: The data indicating that proPC2 is an ERAD substrate is convincing, but the evidence that accumulation of proPC2 and reduction in PC2 activity

due to ERAD disruption is the direct cause of loss of glucagon in these models would benefit from additional supporting data or at least explanation, for example:

(a) If ERAD knockout is causing significant PC2 activity loss, an increased accumulation of proglucagon should be evident, and demonstrated by western blot.

We first attempted to answer this question using our CRISPR-generated α TC knockout cells but unexpectedly found that proglucagon and glucagon are both reduced in this condition, for reasons that we are still exploring (data not shown). Thus, we performed a new experiment using siRNA to acutely knock down SEL1L or HRD1 expression. In this experiment, we observed that acute ERAD dysfunction led to accumulation of both proPC2 and proPC2* (as observed in the CRISPR knockout cells), as well as accumulation of proglucagon. As discussed above, the detection of mature glucagon by Western blot proved technically challenging. These data are now shown in the **new Fig. S7** with accompanying text on pages 9-10.

NEW Figure S7. Control α TC cells were subjected to siRNA-mediated inactivation of Sel1L or Hrd1 (siSel1L or siHrd1, respectively) and labeled for proteins as shown. **a** Representative Western blots and quantification of the ERAD proteins SEL1L and HRD1, proPC2 isoforms (top middle), total PC2 isoforms (bottom middle), and the proPC2 chaperone 7B2. **b** siRNA-treated cells were evaluated for expression of proglucagon-derived peptides using the ABclonal anti-glucagon antibody. As a control, note the lack of proglucagon-to-glucagon processing in α TC cells with CRISPR-mediated inactivation of PC2 (Δ PC2). Each protein band was normalized to the HSP90 loading control, and each data point in the graphs represents an individual replicate. * $P < 0.05$, ** $P < 0.01$, *** $P < 0.001$; NS, not significant ($P > 0.05$); Student's t-test or one-way ANOVA with Šidák post-test.

Revised text, pages 9-10: "To better understand the timing of proPC2 processing defects following ERAD dysfunction, we used an siRNA approach to acutely knock down SEL1L or HRD1 protein levels in control α TC cells (Supplementary Fig. 7). Similar to CRISPR-mediated inactivation of ERAD proteins in α TC cells, we observed accumulation of proPC2 and proPC2* 48 hours after siRNA knockdown of these ERAD proteins (Supplementary Fig. 7a). We also observed accumulation of full-length proglucagon in siSel1L and siHrd1 cells (Supplementary Fig. 7b). Notably, we confirmed the profound accumulation of proglucagon and absence of mature glucagon formation in CRISPR-generated Δ PC2 cells, in which complete inactivation of functional PC2 disrupts proglucagon-to-glucagon conversion, as previously shown⁴². These data further support the role of SEL1L-HRD1 ERAD in proPC2 maturation and subsequent PC2-mediated conversion of proglucagon into the mature glucagon peptide."

(b) The authors could strengthen their conclusions by showing that loss of PC2 activity with PC2 deletion and/or 7B2 inhibitor alters proglucagon production / processing.

The inability of PC2-deficient cells to generate mature glucagon from the proglucagon precursor is well known (Furuta et al., *J Biol Chem* 2001, DOI: 10.1074/jbc.M103362200). Similarly, siRNA-mediated reduction in 7B2 has been shown to disrupt generation of glucagon through disruption of PC2 activity (Helwig et al., *J Biol Chem* 2011, PMC3234932). In our hands, generation of PC2 knockdown α TC cells (using CRISPR) also resulted in accumulation of proglucagon and absence of glucagon, which can be found in the **new Figures S1c and S7b (copied in answers above)**. We have added relevant text for these points on pages 9-10.

(c) As detailed in 1 above, demonstration by western in their models that proglucagon processing is impaired, as immunostaining and ELISAs with available antibodies have unknown or questionable specificity.

We would kindly refer the reviewer to our discussion above regarding the Mercodia assay (Response 1b) and to our new siRNA experiment (outlined in Response 2a).

(d) The decrease in total pancreatic glucagon and proglucagon shown in Fig. 3D does not seem to account for the significant and near proportional loss of alpha cell mass in Fig S3D. Could the loss of glucagon be simply related to loss of alpha cell mass?

The reviewer raises a great point. We agree that ERAD may have additional effects on alpha cell biology, similar to what we reported for beta cells – where ERAD directly controls TGF β signaling but also affects beta cell identity (Shrestha et al. *J Clin Invest* 2020 PMC7324191). While a reduction in alpha cell mass could contribute to the decreased production of glucagon in *Sel1L^{4Gcg}* mice, we believe this is unlikely to be sole cause. Our TEM data shows an ongoing presence of alpha cells with abnormal morphology (i.e., dilated endoplasmic reticulum and fewer secretory granules; Fig. S3a), so it is possible that our immunofluorescence-based alpha cell mass measurement slightly underestimates the total alpha cell population. Our data supports the current model of ERAD affecting proPC2 maturation and glucagon generation, but does not exclude other mechanisms. We have **updated the Discussion on page 14** to reflect these points.

Revised text, page 14: "Although the current work demonstrates a primary role for ERAD in regulating proPC2 maturation in α cells, our data hint at other long-term effects of ER dysfunction in α cell biology. Our CRISPR-generated ERAD-deficient α cell lines unexpectedly showed lower Pcsk2 and Pcsk1 gene expression, which was not observed following short-term inactivation of ERAD using an siRNA approach. These data suggest additional long-term effects on α cell identity that mimic the "dedifferentiation" we observed in ERAD-deficient β cells³⁰. Similarly, we observed decreased α cell mass in older Sel1L ^{Δ Gcg} mice, which may reflect a progressive decline in α cell number, though it is unclear if the immunofluorescence-based measurement underestimated the size of the glucagon-deficient α cell population. It remains challenging to correlate α cell mass with systemic glucagon physiology, as it has been previously shown that near-total ablation of α cells induced by diphtheria toxin dramatically reduced pancreatic glucagon content but had a lesser impact on systemic glucagon levels⁷¹. This suggests a robust capacity for normal α cells (e.g., those that escaped Cre-mediated recombination in our model) to maintain physiologic levels of glucagon secretion. As our female mice started with nearly double the glucagon content of male mice, similar to findings from other groups⁷²⁻⁷⁴, we hypothesize that females have a greater capacity to compensate for α cell dysfunction at younger ages, but (like the males) eventually lose the ability to produce enough glucagon to mount a normal secretory response to hypoglycemia. Thus, there are several potential contributors to the glucagon production defect in Sel1L ^{Δ Gcg} mice beyond the observed misprocessing of proPC2. The specific mechanisms underlying these additional effects remain to be explored."

(e) Why does the functional glucagon defect only manifest in older Sel1L Δ Gcg mice (4 months male; 8 months female)? If a proPC2 processing defect is truly the cause of glucagon dysfunction, would the authors not expect the glucagon defect to be apparent earlier? One interpretation might be due to loss of alpha cells with decreasing with age associated with ERAD disruption. The authors should assess alpha cell population in younger mice. Further, performing cell viability and proliferation assays on the aged mice might be influenced by a survivorship bias.

The reviewer raises another excellent point. It is known that a low number of alpha cells can be sufficient to maintain glucagon blood levels in mice, as near-total ablation of alpha cells induced by diphtheria toxin dramatically reduced pancreatic glucagon content but had a lesser impact on systemic glucagon levels (Thorel et al., *Diabetes* 2011, PMC3198058). This suggests a robust capacity for normal alpha cells (e.g., those that escaped Cre-mediated recombination in our model) to maintain glucagon secretory capacity, particularly as typical stimuli only induce exocytosis of a small portion of an islet's cells granules stores.

We have added data from 11-week-old mice, which show no difference in glucagon secretion in response to insulin-induced hypoglycemia (combined sexes, **new Fig. S2h**) and a statistically significant reduction in pancreatic glucagon content in male but not female Sel1L ^{Δ Gcg} mice (**new Fig. S2i**). As our female mice started with nearly double the glucagon content of male mice (Fig. 3d), similar to findings from other groups (Bonnie-Nielsen, *Metabolism* 1980, DOI 10.1016/0026-0495(80)90014-1; Ling et al., *Virchows Arch* 2001, DOI 10.1007/s004280000374; Jo et al., *J Endocr Soc* 2023, PMC10590649), we hypothesize that females have a greater capacity to compensate for alpha cell dysfunction at younger ages, but (like the males) eventually lose the ability to produce enough glucagon to mount a normal secretory response to hypoglycemia. Thus, we emphasize the difficulty in establishing a precise correlation between alpha cell mass and/or dysfunction with a systemic glucagon

secretory defect. Together, our data are most consistent with ERAD deficiency limiting the total production of mature glucagon in affected cells, with the imperfect nature of Cre-lox recombination allowing for (temporary) compensation in glucagon secretion *in vivo*. We have added these points to the Discussion on page 14 (text copied in response above).

NEW Figure S2h-i. **h** Serum glucagon values measured *in vivo* 30 minutes after insulin-induced hypoglycemia in combined male and female mice at 11 weeks of age. **i** Acid ethanol-extracted pancreatic glucagon content in 11-week-old mice, with sexes as indicated. Each data point represents one mouse and bars showing mean \pm SEM. * $P < 0.05$, NS, not significant ($P > 0.05$), unpaired two-tailed Student's t test.

(f) Measurement (ELISA) of GLP-1 in ERAD deficient alpha cells would be informative. What happens to Pcsk1 / PC1/3 in ERAD deficient alpha cells? It would be of interest if the defect was specific for PC2 and resulted in increased GLP-1 production, though this is not clear.

We agree that this is an important point. We have added **new data (presented in Fig. S6)** to reinforce the finding that ERAD deficiency has a specific and direct effect on (pro)PC2 maturation in alpha cells, and that any effect of ERAD dysfunction on PC1/3 protein expression and GLP-1 production is less pronounced. This new data is discussed on page 8.

Revised text, page 8: "To explore whether the consequences of ERAD deficiency specifically affected proPC2 maturation and glucagon production in α cells, we next examined expression of the PC1/3 enzyme, which is primarily responsible for generation of GLP-1 from proglucagon. By immunofluorescence, Sel1L^{ΔGcg} α cells showed slightly reduced expression of both PC1/3 and GLP-1 (Supplementary Fig. 6a-b). Unlike proPC2, we did not observe aggregation of PC1/3 under nonreducing conditions on Western blot (Supplementary Fig. 6c). Instead, the lower PC1/3 content was attributed in part to reduced expression of Pcsk1 mRNA in Δ Sel1L α cells (Supplementary Fig. 6d). Pancreatic total GLP-1 content, as measured by immunoassay, was significantly reduced in adult male but not female Sel1L^{ΔGcg} mice (Supplementary Fig. 6e). Thus, ERAD deficiency specifically inhibits maturation of proPC2 in α cells and imparts a lesser effect on PC1/3 expression and GLP-1 production."

NEW Figure S6. a-b Representative islets from *Sel1L^{ΔGcg}* mice and *Sel1L^{fl/fl}* controls, immunolabeled for prohormone convertase 1/3 (PC1/3, **a**; RS20 antibody) or glucagon-like peptide 1 (GLP-1, **b**). Scale bars, 25 μm. Violin plots in **a'**-**b'** show the distribution of immunofluorescence intensity of PC1/3 or GLP-1 in glucagon+ cells from 855-2029 cells with $n = 2-3$ mice per genotype, with median line shown in white and quartile lines in black. **c** Western blot of islet lysates from *Sel1L^{fl/fl}* and *Sel1L^{ΔGcg}* mice prepared under reducing and nonreducing conditions, labeled with antibodies to PC1/3 (Cell Signaling Technology), with demonstration of high molecular weight (HMW) isoforms. Quantification of each isoform is shown in **c'**, with each data point representing a single replicate, shown as mean ± SEM. **d** Quantitative RT-PCR of the *Pcsk1* gene encoding PC1/3 in αTC cells. Each data point represents a single replicate, shown as mean ± SEM. **e** Acid ethanol-extracted pancreatic total GLP-1 content in adult (4-8 month-old) mice. Each data point represents data from one mouse, with mean ± SEM shown. * $P < 0.05$, *** $P < 0.001$; ns, not significant ($P > 0.05$); Student's t-test or one-way ANOVA with Šidák post-test.

(g) Does *Pcsk2* mRNA expression change in the mutants?

Pcsk2 mRNA is not increased in Δ *Sel1L* and Δ *Hrd1* αTC cells. This data can be found in Fig. 5h.

3. Concerns with some figures:

(a) Fig 1C – inclusion of insulin staining in top panel of 1C (control) would enable clearer comparison with the Gcg KO panel below.

As BiP expression is low in normal islet alpha cells, we felt that this experiment should have direct labeling of the alpha cells to confirm their location. Insulin labeling was used in the *Gcg*^{-/-} islets solely to demonstrate the peri-islet location of (pro)glucagon-negative alpha cells in this model. To better visualize the alpha cells in *Gcg*^{-/-} mice, we have added immunofluorescence data using adjacent tissue sections labeled with the alpha cell marker transthyretin in the **new Fig. 1c**.

UPDATED Figure 1c. Representative islets from *Gcg*^{-/-} mice and littermate controls, with adjacent tissues sections immunolabeled for the ER chaperone BiP (left) or the α cell marker transthyretin (right) and either glucagon (GCG, top panels; BMA Biomedicals) or insulin (INS, bottom panels).

(b) representative images in Fig 2 could be more representative: in Fig 2C it is hard to see BiP expression changes when there are so many more alpha cells in the control than in the *Sel1L* KOs; Fig 2D does not convincingly show the slight but significant decrease in proglucagon and glucagon levels in *Sel1L* knockouts reported by the quantification.

We have **updated Fig. 2c-d** with more representative islets that reflect the average intensity of BiP, proglucagon, and (pro)glucagon immunofluorescence:

UPDATED Figure 2c-d. Representative islets from adult *Sel1L^{ΔGcg}* mice and *Sel1L^{fl/fl}* littermate controls, immunolabeled for (pro)glucagon and the ER chaperone BiP (c) or proglucagon (d). Note that the BMA Biomedicals antibody targeting the mature glucagon sequence was found to label full-length proglucagon by Western blot (see Supplementary Fig. 1c), so images are labeled as detecting “(pro)glucagon” accordingly. Scale bars of left panels, 50 μ m. Violin plots show the distribution of immunofluorescence intensity of BiP, OS9, proglucagon, or glucagon in a range of 277-1451 α cells from $n = 3-4$ mice per genotype. *** $P < 0.001$, unpaired two-tailed Student's t test.

(c) The presence of a proPC2 band in the PC2 knockout condition is puzzling and would benefit from explanation.

We thank the reviewer for pointing out a potential source of confusion. We believe this simply reflects an incomplete CRISPR knockout. We have added clarifying text to page 8.

Revised text, page 8: “note that PC2 was significantly reduced but incompletely deleted by this method”

Minor

1. The implication that SEL1 is not changed in type 2 diabetes should be softened since it is based on immunostaining from just one WT and one diabetic donor, or more donors included.

We apologize for any misunderstanding; our point was simply that SEL1L expression was observed in alpha cells from islet donors with and without diabetes. We agree that there is not sufficient data to make a conclusion on changes in expression in diabetes. Our comment on this data can be found on page 5, at the end of the first Results paragraph.

Text, page 5: “SEL1L was expressed in both α and β cells in human islets, from donors with or without diabetes (Fig. 1g).”

2. Please state the species of the homemade proPC2 antibody in the methods section.

We have included this information (antibody generated against mouse proPC2, host species rabbit) in the new Table S1.

3. Quantification of OS9 expression in Fig. 2B in a manner similar to BiP in Fig. 2C to provide would strengthen conclusion that it is upregulated.

We have performed quantification and **added this data to Fig. 2** (see updated figure copied in the response to Major Point 3b above).

Reviewer #2 (Remarks to the Author):

This manuscript by Zhu et al investigates the role of the SEL1L-HRD1 ERAD pathway in islet α cells, demonstrating that it is essential for the proper folding, maturation, and enzymatic activation of proPC2, a key enzyme in glucagon biogenesis. The authors provide evidence that Sel1L and Hrd1 knockout in proglucagon-expressing cells leads to aberrant proPC2 processing, accumulation of a truncated proPC2* species, decreased PC2 activity, and ultimately impaired proglucagon cleavage. These findings reveal a post-translational regulation in α cell biology and open potential therapeutic avenues in metabolic diseases.

The paper is clearly structured: the introduction lays out the need for this study, methods are suitable to test their hypothesis, the data support the main conclusions, and discussions are thorough.

Several mechanistic and contextual points should be addressed to solidify the conclusions.

We also thank this reviewer for their supportive comments and helpful feedback to improve the manuscript.

Major concerns:

1. Mechanism of proPC2* generation

The accumulation of proPC2* in ERAD-deficient α cells is intriguing and raises important questions about its origin. While the proposed C-terminal cleavage is plausible, the exact mechanism by which proPC2* is generated remains unclear. Since proPC2* appears to be a potentially important intermediate with pathological relevance, it would greatly strengthen the study to provide further biochemical insight into how this fragment is formed—whether through a specific protease, misfolding, or another pathway.

We agree that the presence of proPC2* is an unexpected and exciting finding. The fact that proPC2* is increased in control α TC cells treated with the MG132 proteasome inhibitor (Fig. 6d, lanes 1 versus 4) and is also observed in low levels in control *Sel1L^{fl/fl}* islets (**new Fig. S6c**) suggests that proPC2* is actively generated in normal alpha cells but is cleared by proteasomal degradation. We now show that proPC2* is sensitive to deglycosylation by Endo H and PNGase (**new Fig. S5d**),

suggesting that it is an immature protein. Thus, we conclude that proPC2* is an abnormal cleavage product generated in the ER of alpha cells (unlike typical proPC2 cleavage in the secretory granules), and that nascent proPC2* is actively cleared by SEL1L-HRD1 ERAD. We are currently exploring the precise nature of this cleavage event as an independent study and have expanded the Discussion on page 13 as such.

NEW Figure S5d. Western blot of αTC, ΔSel1L, and ΔHrd1 cells before and after treatment with endoglycosidase H (Endo H) or Peptide:N-glycosidase F (PNGase F).

Revised text, page 13: "Our data show that the effect of ERAD on proPC2 maturation is direct and uncoupled from ER stress-related protein misfolding, as chemical ER stressors failed to cause the accumulation of proPC2 or the novel truncated proPC2 protein. Instead, proPC2 and proPC2* are SEL1L-HRD1 ERAD substrates that are susceptible to formation of high molecular weight protein aggregates via aberrant disulfide bonds in the absence of ERAD function. The fact that we could detect proPC2* in control α cells after addition of MG132 and in isolated islets from Sel1L^{fl/fl} mice suggests that proPC2* is generated in normal α cells but is quickly degraded by the proteasome, through SEL1L-HRD1 ERAD. Based on the estimated molecular weight of proPC2*, we predict that cleavage of proPC2 within the P domain leads to the generation of the truncated proPC2* protein; however, the specific mechanism producing proPC2* in the ER remains unclear. The neuroendocrine protein 7B2 is necessary for formation of an activation-competent proPC2 complex that can exit the ER, and in its absence proPC2 is susceptible to misfolding and aggregation in CHO cells⁴¹. However, previous 7B2 structure-function and knockout analyses did not show evidence of proPC2* generation^{41,61,62}. Our siRNA and 7B2 overexpression experiments demonstrated that excess 7B2 alone was insufficient to rescue the accumulation of proPC2(*) in ERAD-deficient α cells, which appears to be a rapid consequence of ERAD dysfunction. We speculate that the reduction of available 7B2 cofactor in α cells with chronic ERAD deficiency exacerbates proPC2 aggregation within the ER, possibly by limiting export of nascent proPC2⁴¹. Future studies will be required to clearly define the role of ERAD in the regulation of 7B2 expression and its functional consequences, as well as identifying the specific mechanism by which the aggregation-prone proPC2* peptide is generated."*

2.Role of 7B2 downregulation and feedback loops

The observed downregulation of 7B2 in ERAD-deficient α cells suggests a contributing factor to impaired PC2 maturation. However, this conclusion is largely correlative. The authors are encouraged to assess whether exogenous 7B2 expression can rescue PC2 activity or reduce proPC2 aggregation. This would provide functional support for the 7B2–PC2 axis as a downstream effector of ERAD in α cells.

We thank the reviewer for encouraging a closer investigation into the specific role of 7B2 in ERAD deficiency. The essential role for 7B2 in proPC2 maturation has been well documented (summarized in Muller and Lindberg, *Prog Nucleic Acid Res Mol Biol* 1999, PMID: 10506829). Briefly, 7B2 limits aggregation and prevents premature activation of proPC2, and is required to facilitate proPC2 transport through the

secretory pathway. Here, our initial data showed downregulation of 7B2 in α TC cells following CRISPR-mediated ERAD inactivation (Fig. 6e). To better define the role of 7B2 in ERAD deficiency without any confounding effects of long-term ER dysfunction, we have now investigated 7B2 expression in cells with siRNA-mediated ERAD inactivation. While both siSel1L and siHrd1 cells also demonstrated proPC2 accumulation, they showed an unexpected increase in 7B2 expression (**new Fig. S8a**). Furthermore, overexpression of 7B2 was unable to rescue the accumulation and aggregation of proPC2(*) in Δ Sel1L and Δ Hrd1 α TC cells (**new Fig. S8**), which fits with prior data showing that 7B2 facilitates ER export but not the initial folding of proPC2 (Muller et al., *J Cell Biol* 1997, PMC2141705). Thus, we conclude that proPC2(*) aggregation is a rapid and direct consequence of ERAD deficiency in alpha cells, independent of 7B2 levels. We have added relevant points regarding this data to the Discussion on page 13.

Revised text, page 13: "The neuroendocrine protein 7B2 is necessary for formation of an activation-competent proPC2 complex that can exit the ER, and in its absence proPC2 is susceptible to misfolding and aggregation in CHO cells⁴¹. However, previous 7B2 structure-function and knockout analyses did not show evidence of proPC2 generation^{41,61,62}. Our siRNA and 7B2 overexpression experiments demonstrated that excess 7B2 alone was insufficient to rescue the accumulation of proPC2(*) in ERAD-deficient α cells, which appears to be a rapid consequence of ERAD dysfunction. We speculate that the reduction of available 7B2 cofactor in α cells with chronic ERAD deficiency exacerbates proPC2 aggregation within the ER, possibly by limiting export of nascent proPC2⁴¹. Future studies will be required to clearly define the role of ERAD in the regulation of 7B2 expression and its functional consequences, as well as identifying the specific mechanism by which the aggregation-prone proPC2* peptide is generated."*

NEW Figure S8. a Representative Western blots and quantification of 7B2 and proPC2* in control and CRISPR-treated Δ Sel1L and Δ Hrd1 cells that were transfected with a plasmid expressing rat 7B2. **b** Cells without and with 7B2 overexpression were evaluated under reducing and non-reducing conditions to evaluate for high molecular weight (HMW) forms of proPC2(*). Each protein band was normalized to HSP90 or GAPDH, and each data point in the graphs represents an individual replicate. * $P < 0.05$, ** $P < 0.01$, *** $P < 0.001$; only significant results are shown. One-way ANOVA with Šidák post-test.

3. Rule out the possible off-target effects of the Gcg-Cre model in NTS

Given the use of a Gcg-Cre model, the potential for off-target recombination in L-cells and the central nervous system (e.g., NTS) must be considered carefully. While the authors reference prior studies and the absence of metabolic phenotypes, additional lineage tracing or tissue-specific expression data would help rule out confounding effects.

We agree that these are important considerations, especially with the expectation that these knock-in Gcg-iCre mice theoretically target recombination in all proglucagon-expressing cells. As shown in Fig. S2a-d, we were unable to observe evidence of Cre-mediated recombination in intestinal L-cells, with no detectable YFP reporter expression and no significant changes in BiP (an anticipated consequence of ERAD deficiency). In addition to the finding that plasma and distal colon tissue levels of GLP-1 were unchanged (Fig. S2f-g), we strongly believe that any effect of SEL1L-

HRD1 ERAD deficiency in L-cells has a minimal impact on systemic glucose physiology.

We have since examined brain sections from *Sei1L^{ΔGcg}* mice and littermate (Cre negative) controls. As shown in **Response Fig. 1**, we observed YFP reporter expression in (pro)glucagon+ cells of the NTS, but without increased expression of the ER marker KDEL. The lack of neurobehavioral or systemic metabolic phenotypes strongly supports that the primary phenotype observed in this mouse model arises from ERAD deficiency in islet alpha cells. We have added relevant Discussion points on page 12.

Response Figure 1. Immunohistochemistry of brain cryosections highlighting cells within the nucleus of the solitary tract (NTS) labeled with antibodies to GFP (detecting the YFP reporter), (pro)glucagon, and the ER marker KDEL.

Revised text, page 12: "It is also unlikely that this model has impactful ERAD dysfunction in neurons given the lack of body weight changes or systemic glucose dysregulation as observed with manipulation of proglucagon expression in the CNS⁵⁹ and the absence of neurologic dysfunction that we have observed in models inducing ERAD dysfunction in the brain^{27,60}. Given that the mature glucagon peptide is derived nearly exclusively from islet α cells (excluding observations from pancreatectomized mammals), and given our primary finding of ERAD deficiency causing a reduction in pancreatic and circulating glucagon levels, we believe that any impact of Cre-mediated recombination in L-cells or neurons in this model is minimal."

Recommendation: Major Revision